DOI: 10.1038/s41467-018-06994-5　　**OPEN**

# High-power hybrid biofuel cells using layer-by-layer assembled glucose oxidase-coated metallic cotton fibers

Cheong Hoon Kwon [1], Yongmin Ko[1,2], Dongyeeb Shin[1], Minseong Kwon[1], Jinho Park[2], Wan Ki Bae[3], Seung Woo Lee [2] & Jinhan Cho [1]

Electrical communication between an enzyme and an electrode is one of the most important factors in determining the performance of biofuel cells. Here, we introduce a glucose oxidase-coated metallic cotton fiber-based hybrid biofuel cell with efficient electrical communication between the anodic enzyme and the conductive support. Gold nanoparticles are layer-by-layer assembled with small organic linkers onto cotton fibers to form metallic cotton fibers with extremely high conductivity ($>2.1 \times 10^4$ S cm$^{-1}$), and are used as an enzyme-free cathode as well as a conductive support for the enzymatic anode. For preparation of the anode, the glucose oxidase is sequentially layer-by-layer-assembled with the same linkers onto the metallic cotton fibers. The resulting biofuel cells exhibit a remarkable power density of 3.7 mW cm$^{-2}$, significantly outperforming conventional biofuel cells. Our strategy to promote charge transfer through electrodes can provide an important tool to improve the performance of biofuel cells.

[1] Department of Chemical and Biological Engineering, Korea University, 145 Anam-ro, Seongbuk-gu, Seoul 02841, Republic of Korea. [2] The George W. Woodruff School of Mechanical Engineering, Georgia Institute of Technology, Atlanta, GA 30332, USA. [3] SKKU Advanced Institute of Nano Technology (SAINT), Sungkyunkwan University, Seobu-ro, Jangan-gu, Suwon-si, Gyeong gi-do 16419, Republic of Korea. These authors contributed equally: Cheong Hoon Kwon, Yongmin Ko. Correspondence and requests for materials should be addressed to S.W.L. (email: seung.lee@me.gatech.edu) or to J.C. (email: jinhan71@korea.ac.kr)

Biofuel cells (BFCs)[1] are thought to be one of the most attractive power sources for implantable and microscale biomedical devices because of their biocompatibility and operation at mild temperatures and near-neutral pH[2–4]. Despite these advantages, the relatively low power output of BFCs compared to other power sources, resulting from slow electron transfer, poor utilization efficiency of the enzyme, and limited mass transport, has restricted their practical applications. In particular, poor electrical communication between the enzyme and the conductive support has a critical effect on the power output of BFCs[5,6]. To overcome this problem, redox mediators between the enzyme and the conductive support have been introduced to improve the electrical communication between the two, thereby enhancing the power performance of the cell. To the best of our knowledge, the best power performance reported to date for these mediated electron-transfer-based BFCs (MET-BFCs) is 2.18 mW cm$^{-2}$ [7]. However, the incorporation of a redox mediator in BFCs can lead to several issues, such as toxicity, instability, the complex synthesis of redox mediators, and a reduced open-circuit voltage. Thus, recent research efforts have focused on the development of direct electron transfer-based BFCs (DET-BFCs) with improved performance[8].

A variety of conductive carbon materials (e.g., carbon nanotubes (CNTs)[9–11], graphene[12], and porous carbon[13]) with large surface areas, 3D porous structures for facile mass transport, and high electrical conductivities (<300 S cm$^{-1}$) have been proposed as supports having favorable electrical communication with the enzyme layer. For example, Zebda et al. demonstrate that DET-BFCs based on CNT supports exhibited a significantly higher power output of 1.25 mW cm$^{-2}$ compared to previously reported DET-BFCs fabricated by solution drop casting and high mechanical compression[5]. Despite all the merits of the carbon substrates for the BFCs, their biosafety for long-term operation in the in vivo systems is still controversial due to the cellular toxicity nature[14,15]. Furthermore, in most cases, the enzymes have been directly deposited onto these carbon supports through conventional physical adsorption, which has limitations in terms of the precise control of the interfacial distance, conformation, and stability between the enzyme and the conductive support, resulting in poor utilization of the enzymes for catalyzing electrochemical reactions.

With the advances of the anode for the glucose oxidation reaction through glucose oxidase (GOx), the proper design of the cathode to improve the oxygen reduction reaction (ORR) activity is also crucial to obtain the high-power density and efficiency. In this regard, the cathodic enzyme, such as laccase or bilirubin oxidase, have been intensively studied due to their excellent ORR activity[5,16]. However, the catalytic activity and stability are strongly dependent on the physiological conditions of the mammal's body fluid such as pH, temperature, and chlorine concentration[17–20]. As an alternative, catalytic metal nanoparticles (NPs) such as Pt, Pd, and Au with high ORR activity have been considered as the competent candidates to the abiotic ORR catalysts[21,22]. Despite the improved performance, conventional methods of depositing NPs onto the substrates, such as the mixing of NPs with carbon-substrates[23] or chemical reduction of the metal precursor, etc.[22,24] have poor controls over the structures of NPs on the substrates. In addition, chemical reduction methods often cause particle agglomeration between adjacent NPs with the reduced electrochemical surface area, which hinders high mass loading of NPs on the substrates[25]. Therefore, it is ideal to incorporate the catalytic NPs onto the substrates in a controlled, 3D spatial arrangement without agglomeration, which can facilitate electron and ion transport for the efficient electrocatalytic reaction.

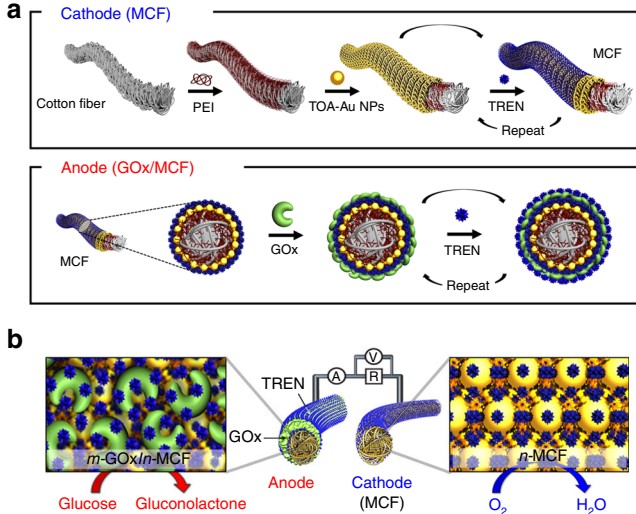

**Fig. 1** Illustration of the metallic cotton fiber electrode-based biofuel cell. **a** Preparation of the metallic cotton fiber (MCF)-based cathode and the glucose oxidase (GOx)/MCF-based anode using small-molecule ligand-induced layer-by-layer assembly. At the first stage, bare cotton fiber was coated with poly(ethylenimine) (PEI, $M_w$ ~800). **b** Redox process for an MCF-biofuel cell (BFC) composed of a cathode and an anode. n-MCF: (tetraoctylammonium bromide-stabilized Au nanoparticle (TOA-Au NP)/tris-(2-aminoethyl)amine (TREN))$_n$, m-GOx: GOx/tris-(2-aminoethyl) amine (TREN)$_m$. The redox process of MCF-BFC, the resistance (R) of the applied external load and the location of voltage (V) and current (A) measuring devices. At the anode, glucose is oxidized to gluconolactone, where the electrons are transferred from the GOx to MCF. At the cathode, electrons are transferred from MCF through the electrolyte with oxygen reduction reaction (ORR)

Layer-by-layer (LbL) assembly methods offer effective opportunities to prepare enzymatic or conductive multilayer films with tailored thickness, functionalities, and composition on substrates of various sizes and shapes utilizing complementary interactions[26–38]. However, as bulky insulating polyelectrolytes are generally LbL-assembled with the desired functional components (e.g., metal NPs, enzymes, functional polymers[39]) to build up multilayers, the mutual electron transfer is significantly suppressed due to the increased contact resistance[40]. Therefore, the ideal LbL assembly for BFC applications should replace bulky polymer electrolytes with small-molecule linkers that can facilitate the electrical communication between the active materials as well as enable the efficient assembly of the catalysts with a controlled structure and loading amount.

Here, we introduce a highly porous metallic cotton fiber-based BFC (MCF-BFC) fabricated using small-molecule linker-induced LbL assembly of the active catalysts, which significantly boosts the DET rate between the enzymes and the conductive supports (Fig. 1). In particular, the Au NPs are selected to fabricate the MCF because of their several benefits in electrochemical biomedical fields[41], such as the superior biocompatibility[42], more facile surface modification[43], and higher electrical conductivity. In this case, the MCFs act as not only the conductive host for depositing the anodic enzyme materials, but also the electrocatalytic cathode for ORR. More importantly, it is worth noting that the loading amount of the metal NPs in the MCF can be easily increased forming the 3D porous structure according to the increase of the bilayer number, and thereby achieving both the higher electrical conductivity and the improved ORR activity without cathodic enzymes. As a result, our hybrid MCF-BFC provides a remarkable power density of 3.7 mW cm$^{-2}$ using a fixed external resistance

without the aid of redox mediators, significantly exceeding the power density of the previously reported conventional DET- and MET-BFCs.

For the fabrication of the MCF, instead of bulky polymers such as polyelectrolytes, we employ an amine-functionalized small molecule, tris-(2-aminoethyl)amine (TREN), that can electrostatically interact with GOx as well as covalently interact with Au NPs. First, LbL assembly of TREN and tetraoctylammonium bromide-stabilized Au NPs (TOA-Au NPs) was carried out on cotton fibers. In this case, the amine groups of TREN can covalently adsorb on the surface of Au NPs due to its stronger affinity than the ammonium group of TOA. Then, negatively charged GOx was electrostatically LbL-assembled onto the TOA-Au NP/ TREN multilayer-coated cotton fibers using positively charged TREN in water. Based on this unique LbL assembly of conductive NPs and enzyme materials through the same small-molecule linker allowing different interactions, we fabricate entirely different MCF electrodes that have extremely high electrical conductivity ($>2.1 \times 10^4$ S cm$^{-1}$) and yet still maintain the intrinsic 3D porous structure of cotton fibers with conformal coatings of the active layers. With its extremely high electrical conductivity and highly porous 3D structure, MCF can facilitate charge transfer to the conformally coated enzyme layers as well as the mass transport of glucose (for oxidation by GOx) and oxygen (for reduction by Au NP) through the electrodes, thus enabling effective electrocatalytic reactions. Furthermore, the robust covalent bonding between the Au NPs and TREN and the electrostatic bonding between GOx and TREN increase the operational stability of the MCF-BFCs. Given that these unique characteristics of MCFs play a decisive role in enhancing the power output of hybrid BFCs, this approach can contribute to the design of other high-performance electrodes for various electrochemical applications, including other energy storage devices and sensors.

## Results

**Performance of metallic cotton fiber cathodes.** For the preparation of the MCF electrode, TOA-Au NPs with a diameter of 8 nm[43] were LbL-assembled with TREN onto a cotton fiber composed of multiple hydrophilic microfibrils (cellulose fibers containing hydroxy groups). The TOA-Au NPs were uniformly and densely coated onto all the microfibrils that make up the cotton fiber (~200 μm diameter) without disrupting the inherently porous structure (Fig. 2a, b and Supplementary Fig. 1). This was confirmed by field-emission scanning electron microscopy (FE-SEM), energy-dispersive X-ray spectroscopy (EDS) mapping, and topographic atomic force microscopy (AFM) (Fig. 2c and Supplementary Fig. 2). It is worth noting that the amount of (TOA-Au NP/TREN)$_n$ multilayers loaded onto the porous cotton fibers (8.5 mg cm$^{-2}$ for $n = 20$) was more than ten times higher than that loaded onto nonporous nylon fiber, which has the same dimension (~200-μm-diameter) as a cotton fiber (0.8 mg cm$^{-2}$ for $n = 20$) (Fig. 2d and Supplementary Fig. 3). The electrical conductivity and resistivity of the MCF showed strong correlations with the number of bilayers; a high conductivity of ~3.5×10$^3$ S cm$^{-1}$ and a low resistivity of ~1.2×10$^{-4}$ Ω cm were measured for (TOA-Au NP/TREN)$_{20}$ multilayers (Fig. 2e). This correlation indicates that the electrical properties of the MCFs with higher loading are closer to those of bulk Au wire (electrical conductivity ~4.1×10$^5$ S cm$^{-1}$, resistivity ~2.1×10$^{-6}$ Ω cm). It should be noted that the conductivity of bare CNT fibers does not exceed 600 S cm$^{-1}$ without additional post-treatment[44,45]. Furthermore, the formed MCF retains its intrinsic flexibility even though Au NPs are densely loaded onto the bare cotton fiber (Supplementary Fig. 4).

The high electrical conductivity of MCFs is caused by the unique adsorption mechanism, in which the bulky and hydrophobic TOA ligands of the TOA-Au NPs are sequentially replaced by small TREN ligands containing amine (NH$_2$) groups during the LbL assembly. As confirmed by Fourier transform infrared (FTIR) spectroscopy (Fig. 2f and Supplementary Fig. 5), the first deposition of TOA-Au NPs onto a poly(ethylenimine) (PEI, $M_w$ ~800)-coated Si wafer (i.e., 0.5 bilayer) generated the characteristic peaks of TOA ligands (C−H stretching vibrations at 2928 and 2856 cm$^{-1}$) and the amine groups of PEI (N−H bending at 1654 and 1554 cm$^{-1}$). The subsequent adsorption of TREN onto the outermost TOA-Au NP layer (i.e., 1 bilayer) eliminated the C−H stretching peaks corresponding to the TOA ligands with long alkyl chains, while the N−H bending peaks were intensified by TREN adsorption. As a result, this sequential ligand exchange reaction allows TREN to act as a new molecular linker between adjacent Au NPs instead of the bulky TOA ligands, which minimizes the distance and contact resistance between the Au NPs.

We also investigated the electrochemical properties of the (TOA-Au NP/TREN)$_n$-coated cotton fiber electrodes (hereafter abbreviated as $n$-MCF) as a function of the number of bilayers ($n$) at a scan rate of 5 mV s$^{-1}$ using a three-electrode cell configuration in 20 mmol L$^{-1}$ phosphate-buffered saline (PBS) solution under ambient conditions. Cyclic voltammetry (CV) scans showed a gradual increase in cathodic current densities when the number of bilayers ($n$) was increased from 5 to 20 due to the increased active surface area from the introduced Au NPs[46] (Fig. 2g) (please note that the power density of MCF-based hybrid BFCs was evaluated by measuring the current output applying a fixed external resistance to eliminate the capacitive and other parasite currents). Additionally, electrochemical impedance spectroscopy (EIS), obtained from the fitting of the Randle's equivalent circuit, also demonstrated the improved electrical conductivity of 20-MCF, with significantly decreased equivalent series resistance (ESR) of 30 Ω (at 1 kHz) and charge transfer resistance ($R_{ct}$) of 4 Ω relative to those of the 5-MCF electrode (ESR ~1346 Ω and $R_{ct}$ ~303 Ω; Fig. 2h, Supplementary Fig. 6a, and Supplementary Note 1). These results clearly show that the electrochemical properties of MCF electrodes are governed by the Au NP loading[47], which can be further improved by increasing the number of bilayers. In addition, the very high knee frequency of 20-MCF (119.4 Hz) compared to 5-MCF (10.9 Hz) indicates better mass transport for the highly porous structure with sufficient Au NP coverage (Supplementary Fig. 6b). Therefore, an $n$-MCF electrode can provide bilayer-dependent electrocatalytic activity toward the ORR as a cathode, and it can also be used as a host for the anode (ultimately, for a complete hybrid BFC).

To confirm these possibilities, we further investigated the electrochemical cathodic properties of MCFs using CV in oxygen-free (N$_2$), ambient, and oxygen-rich PBS solutions (Supplementary Fig. 7a, b). The measured cathodic current densities of the 20-MCF electrode in ambient and O$_2$ conditions at a potential of −0.6 V were found to be −17.1 and −27.9 mA cm$^{-2}$, respectively. The normalized current density levels were obtained by subtracting the areal cathodic current density level measured at a specific potential of −0.6 V in oxygen-free condition from the areal cathodic current density level measured at the same potential in ambient or oxygen-rich condition. On the basis of −7.1 mA cm$^{-2}$ in oxygen-free N$_2$ condition, the normalized cathodic current densities at −0.6 V were −10 mA cm$^{-2}$ (with an ESR of 33 Ω) under ambient condition, and −20.7 mA cm$^{-2}$ (with an ESR of 29 Ω) under O$_2$ condition (Supplementary Fig. 7c, d).

In addition, it should be noted that the measured cathodic current density (−17.1 mA cm$^{-2}$) for the 20-MCF electrode

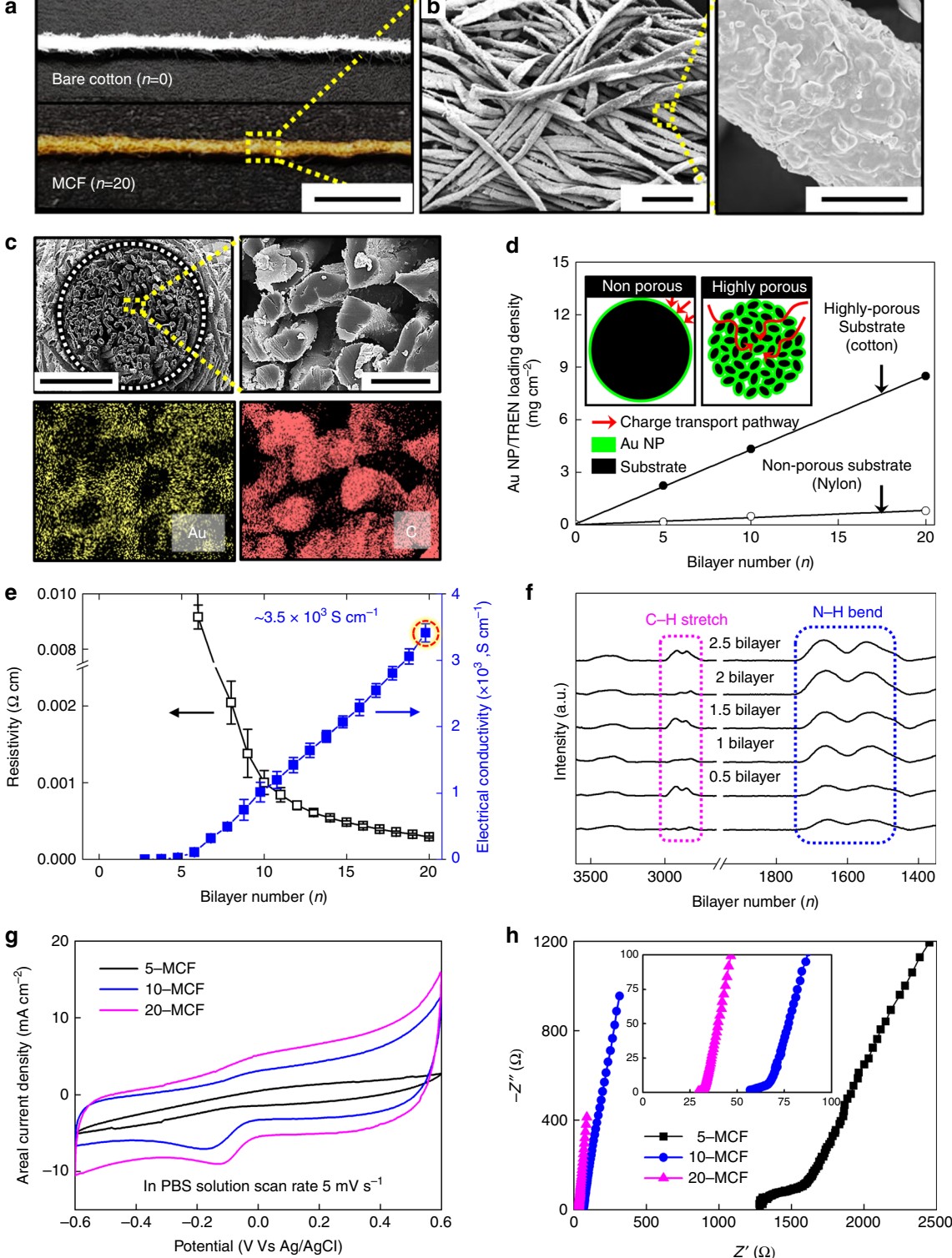

**Fig. 2** Characterization of the metallic cotton fiber electrode. **a** Photographic images of bare cotton fiber (i.e. 0-MCF, metallic cotton fibers is abbreviated as MCF) and 20-bilayered metallic cotton fibers (20-MCF) (scale bar, 1 mm). **b** Field-emission scanning electron microscopy (FE-SEM) images of the 20-MCF (scale bar, 50 μm (left) and 5 μm (right)). **c** Cross-sectional FE-SEM, and energy-dispersive X-ray spectroscopy (EDS) mapping images of the 20-MCF electrode with a diameter of ~200 μm (scale bar, 100 μm (left) and 20 μm (right)). **d** Change in the areal loading amount of the (tetraoctylammonium bromide-stabilized Au nanoparticle (TOA-Au NP)/tris-(2-aminoethyl)amine (TREN))$_n$ multilayers adsorbed onto porous cotton fibers and nonporous nylon fibers with the same diameter (~200 μm) as the number of bilayers ($n$) increases. **e** Resistivity and electrical conductivity of $n$-MCFs as a function of $n$. The error bars show the standard deviation from the mean value of electrical conductivities for 3−5 independent experiments. **f** Fourier transform infrared (FTIR) spectra of the (TOA-Au NP/TREN)$_n$ multilayers as a function of $n$. **g** CV curves of $n$-bilayered metallic cotton fibers ($n$-MCF) at a scan rate of 5 mV s$^{-1}$ in a phosphate-buffered saline (PBS) solution. **h** Nyquist plots of $n$-MCFs in the frequency range 0.2 Hz to 100 kHz. Inset: Nyquist plots magnified in the high-frequency range. The equivalent series resistance (ESR) values of $n$ = 5, 10, and 20 were 1346, 60, and 30 Ω, respectively

outperforms the current density level ($-3.0$ mA cm$^{-2}$) of nonporous bulk Au wire with the same diameter in ambient conditions (Supplementary Fig. 8). This remarkable cathodic behavior of an MCF electrode is caused by the extremely large surface area of the electrocatalytic Au NP-based MCF, which arises from the microporous structure of the cotton fibrils as well as the nanoporous structure of the Au NP multilayers.

**Performance of glucose oxidase/metallic cotton fiber anodes.** We prepared MCF-based anodes through the additional deposition of electrostatically LbL-assembled (anionic GOx/cationic TREN) multilayers onto 20-MCF in PBS solution. In this process, GOx, with an isoelectric point of pH 4.3, has a net negative charge in pH 7.4 PBS solution, which can induce the electrostatic adsorption of positively charged TREN molecules (having protonated amine groups NH$_3^+$ at pH 7.4). The LbL-assembled GOx can catalyze the oxidation of glucose to gluconolactone, and the electrons generated can be directly transferred from the enzyme to the MCF electrode (Fig. 3a). First, the LbL assembly behavior of the (GOx/TREN)$_m$ (where $m$ is the number of bilayers) multilayers on (TOA-Au NP/TREN)-coated quartz glass was investigated by ultraviolet-visible (UV−Vis) spectroscopy (Fig. 3b and Supplementary Fig. 9). In this case, we observed linear hyperchromism at 277 nm associated with the oxidized flavin cofactors of GOx as a function of $m$, implying the uniform and sequential formation of GOx/TREN multilayers (the inset of Fig. 3b). This growth behavior was further quantitatively monitored using a

quartz crystal microbalance (QCM). The increase in frequency changes ($-\Delta F$) or the decrease in mass changes ($\Delta M$) after deposition of each TREN layer can be attributed to the partial desorption of weakly adsorbed GOx during the adsorption of TREN (Supplementary Fig. 10). Additionally, the film thickness of the formed (GOx/TREN)$_m$ multilayers, which are sequentially deposited with small-molecule linkers instead of conventional bulky polyelectrolytes[35–37], was estimated to be ~9 ± 3 nm and ~13 ± 5 nm for 20 and 30 bilayers, respectively (Supplementary Fig. 11). Considering the dimension of GOx (6.0 nm × 5.2 nm × 7.7 nm)[48], these multilayer thicknesses indicate that (GOx/TREN)$_{20}$ and (GOx/TREN)$_{30}$ are adsorbed on the substrates like a single GOx layer and double GOx layers, respectively. As a result, the adsorption conformation of thin and uniform (GOx/TREN)$_m$ multilayers can facilitate the electron transfer through the enzyme layers.

We also examined the glucose electrooxidation activity of the anode as a function of $m$ in PBS solution containing glucose (300 mmol L$^{-1}$) at a scan rate of 5 mV s$^{-1}$ (Fig. 3c and Supplementary Fig. 12). The CVs of the (GOx/TREN)$_m$ multilayer-coated 20-MCF electrode (hereafter abbreviated as $m$-GOx/20-MCF) showed that the overall current density of the electrode steadily increased with increasing number of bilayers ($m$) from 0 to 30. This bilayer ($m$)-dependent anodic performance can be explained by different adsorption behavior of the GOx layer on the inner and outer surface of the MCFs. In other words, in the case of the GOx/TREN multilayers onto the interior of highly porous MCFs, it is considered that their thicknesses will be much thinner with

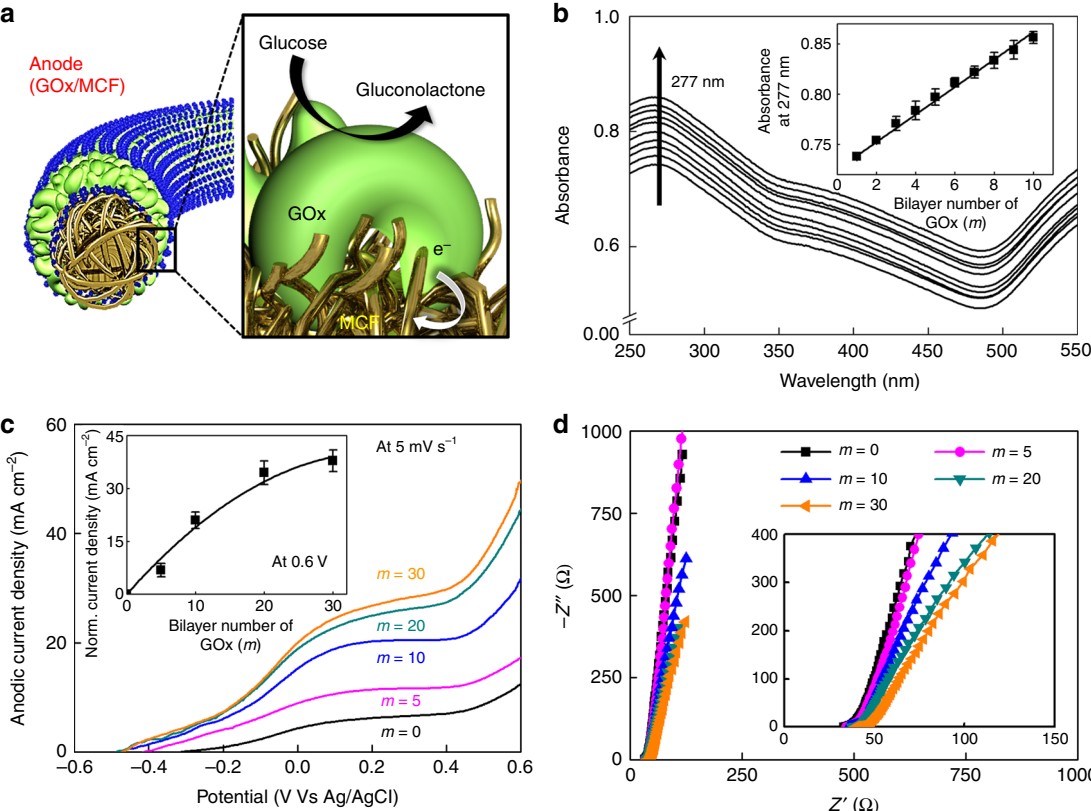

**Fig. 3** Electrochemical properties of biofuel cell anode. **a** Illustration of the anode fibers composed of $m$-bilayered glucose oxidase ($m$-GOx)/20-bilayered metallic cotton fibers (20-MCF). **b** Ultraviolet-visible (UV−Vis) spectra of the GOx/tris-(2-aminoethyl)amine (TREN)$_m$ multilayers. Inset: hyperchromicity of the multilayers at 277 nm. **c** Anodic current density curves of the (GOx/TREN)$_m$/20-MCF electrode. Inset: the change in the normalized anodic current density levels at +0.6 V; here, the normalized level is defined as the current density level for which the anodic current density level of 20-MCF ($m = 0$, black curve) is subtracted from the anodic current density level of $m$-GOx/20-MCF at the specific potential of +0.6 V. The error bars show the standard deviation from the mean value of current densities for 3−5 independent experiments. **d** Nyquist curves for (GOx/TREN)$_m$/20-MCF

going from the outer surface to the center region of MCFs because the repetitive deposition of bulky GOx layers can block up the porous structure of MCFs. Therefore, the overall anodic current density of MCF-based anode electrode can be gradually increased up to the thickness of GOx/TREN multilayers allowing efficient electron tunneling. The anodic current density level measured at +0.6 V approached a maximum value of ~50 mA cm$^{-2}$ (the normalized anodic current density level at +0.6 V was ~40 mA cm$^{-2}$, the inset of Fig. 3c) after the deposition of 30 bilayers.

In this case, the 30-GOx/20-MCF anode maintained ~97.5% of its initial current density after continuous operation for 1 h (herein, the initial current density of the anode electrode was obtained after pretraining for 1 day in 300 mmol L$^{-1}$ glucose containing PBS buffer condition), indicating the excellent electrode stability caused by the precisely controlled deposition of GOx using TREN (Supplementary Fig. 13). Conversely, the ESR value slightly increased with increasing $m$, indicating an increase in the internal resistance, corresponding to the increased loading amount of insulating GOx/TREN multilayers onto 20-MCF (Fig. 3d). However, the ESR value of the MCF-based anode electrode is much lower than that of the nonporous electrode (i.e. Au-coated Si wafer-based anode) (Supplementary Fig. 14) although the loading amount of (GOx/TREN)$_{30}$ multilayers onto MCF electrode is 185 times higher than that of (GOx/TREN)$_{30}$ multilayers onto nonporous electrode (i.e., ~860 μg cm$^{-2}$ for MCF-based anode and ~4.65 μg cm$^{-2}$ for nonporous substrate-based anode at 30 bilayers), which indicate the effective electrode conformation of the MCF-based anode for facile electrochemical kinetics.

We also observed that the electron transfer kinetics at the interface of the electrode are significantly influenced by a kind of outermost layer of GOx/TREN multilayers (Supplementary Fig. 15). More specifically, OH-functionalized glucose has a higher affinity (mainly, hydrogen-bonding interaction) with amine-functionalized TREN than negatively charged GOx at pH 7.4 (due to the presence of carboxylate ion (COO$^-$) groups of GOx). In line with this high affinity, glucose could be easily diffused into the outermost TREN-coated multilayer films, and therefore the outermost TREN-coated electrode exhibited much lower electron transfer resistance than the outermost GOx-coated electrodes in spite of the increase of layer number. These similar phenomena were also demonstrated by Willner group[49]. Considering that all the BFCs shown in our study are based on the outermost TREN-coated electrodes as well as the highly porous electrodes, the electron transfer resistances of MCF-based electrodes can be significantly lowered compared to those of other nonporous electrodes.

Furthermore, we investigated the oxidation activity of the (GOx/TREN)$_{30}$ electrode when the glucose concentration in the PBS solution was increased from 0 to 300 mmol L$^{-1}$ in PBS solution (Supplementary Fig. 16). The anodic current density measured at +0.6 V reached a saturated level at 300 mmol L$^{-1}$ glucose, revealing the Michaelis–Menten-type behavior[50] of the GOx (the inset of Supplementary Fig. 16b). In this case, the normalized anodic current density of the electrode increased to ~40 mA cm$^{-2}$ at a glucose concentration of 300 mmol L$^{-1}$, showing a high electrooxidation reaction involving GOx on the MCF without additional redox mediators. Additionally, a linear dependence of the normalized current densities on the low glucose concentration ranging from 0 to 300 mmol L$^{-1}$ suggests the possibility that the (GOx/TREN)$_m$–coated electrodes can be also used as in vivo glucose sensors (Supplementary Fig. 16d). We also found that the Au NPs on cotton fibers can catalyze the electrooxidation of glucose in aqueous electrolytes[51], generating the anodic current without GOx. In the case of GOx-free MCFs,

the anodic current density was gradually increased with increasing the concentration of glucose in PBS, which corresponds to approximately 22% of the anodic performance of 30-GOx/20-MCF (Supplementary Fig. 17). These results indicate that the highly porous MCFs can serve as a promising host for effective loading of the enzyme catalyst (GOx) as well as the co-catalyst for the oxidation of glucose. Particularly, it should be here noted that these high electrocatalytic activities of the highly porous (GOx/TREN)$_m$ electrodes are in stark contrast to those of nonporous substrate-based electrodes. When the (GOx/TREN)$_m$ multilayers were deposited onto (TOA-Au NP/TREN)$_{20}$ multilayer-coated Si wafers, the (GOx/TREN)$_{10}$-coated flat substrate anode in a 300 mmol L$^{-1}$ glucose solution exhibited a maximum normalized current density level of ~0.01 mA cm$^{-2}$ at +0.6 V (Supplementary Fig. 18).

In addition to the highly porous structure, the excellent electrical conductivity of MCFs also plays an important role on the anode performance. To verify the effect of the electrical conductivity on the catalytic properties of anode electrodes, we further investigated the electrochemical properties for two different kinds of MCFs with different electrical conductivities. For demonstrating this issue, the (TOA-Au NP/PEI)$_{20}$-coated cotton fiber (i.e. PEI-MCF) was prepared as a comparison target of the (TOA-Au NP/TREN)$_{20}$-coated cotton fiber (i.e. TREN-based MCFs). In this case, the PEI-based MCF showed an extremely low electrical conductivity of ~0.02 S cm$^{-1}$ compared to that of the TREN-based MCF (~3.5×10$^3$ S cm$^{-1}$) due to the use of bulky insulating PEI ($M_w$=25,000) despite their similar physical characteristics such as total thickness and surface morphology (Supplementary Fig. 19). Then, the (GOx/TREN)$_{20}$ multilayers were sequentially deposited onto these MCFs. Although the same layer number (or mass) of GOx was immobilized onto each MCF (i.e. PEI- and TREN-based MCFs), the electrooxidation response of the PEI-based MCF anode was significantly low compared to that of the TREN-based MCF (Supplementary Fig. 20), which mainly originated from the poor electron and mass transfer efficiency of PEI-based MCF[52]. These results clearly imply that the highly conductive and porous TREN-based MCF can be used as an effective host for GOx.

To further investigate the catalytic kinetics of 30-GOx/20-MCF (hereafter, 20-MCF means TREN-based 20-MCF) anode, the apparent heterogeneous electron transfer rate ($K_s$) was estimated using the Laviron method[53] for a surface-controlled electrochemical system (Supplementary Fig. 21). In this case, $K_s$ of the electrode was obtained by using the plot of potential ($E-E^0$) vs. scan rate (log $\nu$), which was recorded from the CV data. As a result, the 30-GOx/20-MCF anode exhibited a relatively large $K_s$ value of 6.0 ± 0.1 s$^{-1}$ and a small redox peak potential separation ($\Delta E_p$) of 60 mV at a scan rate of 100 mV s$^{-1}$. Here, small $\Delta E_p$ (60 mV at a scan rate of 100 mV s$^{-1}$) indicates fast heterogeneous electron transfer process, and the reaction is not a diffusion-controlled process but a surface-controlled one, as expected for immobilized system (more specifically, in the case of the surface-controlled system, $\Delta E_p$ is less than 200 mV[54]). Additionally, a large $k_s$ of 6.0 ± 0.1 s$^{-1}$ obtained from 30-GOx/20-MCF suggests that the electrode is significantly effective in facilitating the direct electron transfer of GOx[55–57]. As a result, the formed 30-GOx/20-MCF electrode provides efficient catalytic kinetics with smaller $\Delta E_p$ and larger $k_s$ (for example, 1.53 s$^{-1}$ for multi-walled carbon nanotubes (MWNT)-modified electrode, 0.3 s$^{-1}$ for single-walled carbon nanotubes (SWNT)-modified electrode, 3.96 s$^{-1}$ for the porous TiO$_2$ electrode, and 6.28 s$^{-1}$ for GOx-carbon nanodots on the glassy carbon electrode)[58].

We also highlight that the use of a small-molecule linker between neighboring enzymes can allow facile charge and/or mass transport through the enzyme layer. For comparison

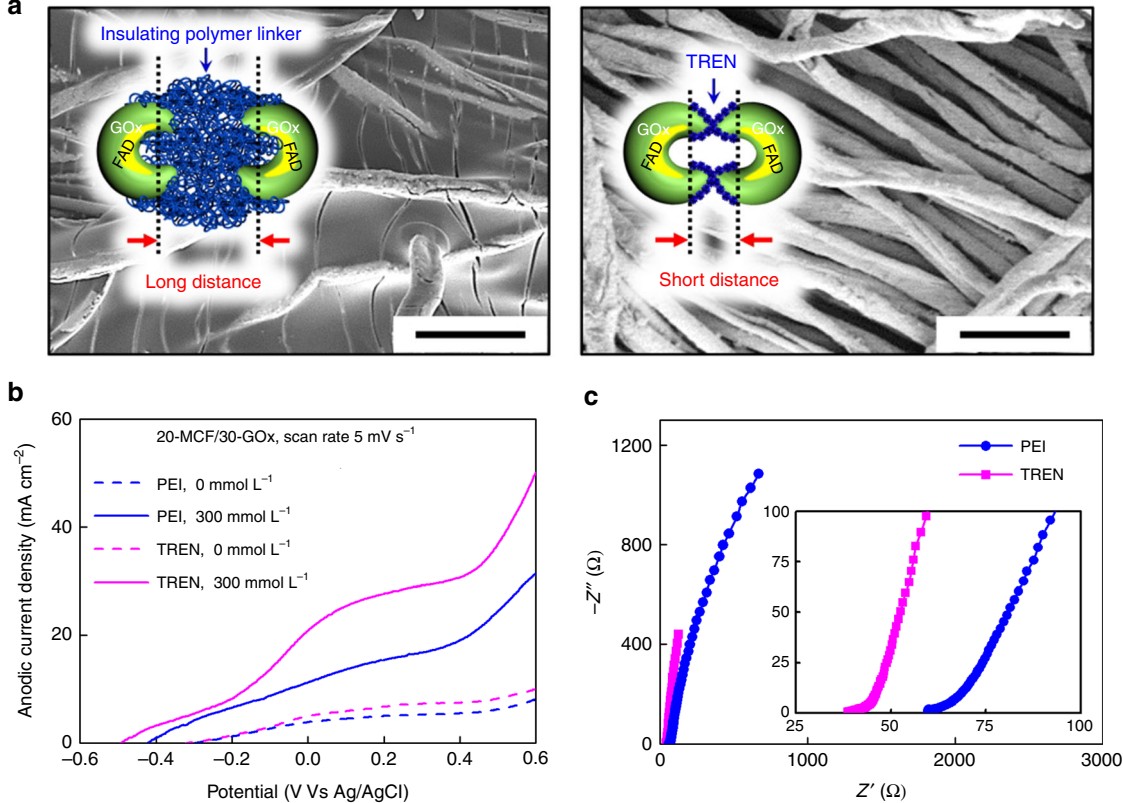

**Fig. 4** Linker type-dependent biofuel cell anodes. **a** Field-emission scanning electron microscopy (FE-SEM) images and illustrated top views of (glucose oxidase (GOx)/poly(ethylenimine) (PEI))$_{30}$ and GOx/tris-(2-aminoethyl)amine (TREN)$_m$ multilayer-coated metallic cotton fibers (MCFs). The scale bars indicate 100 μm. Inset: Schematic diagram showing the difference of the insulating polymer linker (PEI) and small-molecule linker (TREN), which is related to the internal distance (right). In glucose oxidase, flavin adenine dinucleotide (FAD) acts as an electron acceptor. **b** Anodic current density curves of 30-GOx/20-MCF electrodes prepared from (GOx/PEI)$_{30}$ and (GOx/TREN)$_{30}$ in the absence and presence of glucose (300 mmol L$^{-1}$). **c** Nyquist plots for the 30-bilayered GOx/TREN multilayer-coated MCF (i.e., 30-GOx/20-MCF) anodes prepared using PEI and TREN linkers. The equivalent series resistance (ESR) values of the PEI- and TREN-based 30-GOx/20-MCF anodes were 62 and 40 Ω, respectively, in phosphate-buffered saline (PBS) containing 300 mmol L$^{-1}$ glucose

between small-molecule and polymer linkers, GOx was electrostatically LbL-assembled with PEI ($M_w$ ~25,000), a bulky polymer linker, onto 20-MCF for the preparation of (GOx/PEI)$_{30}$/20-MCF electrodes (Fig. 4a). In this case, not only the nanopores but also the micropores in the micro-fibril structure of the MCF-based electrode disappeared with the deposition of (GOx/PEI)$_{30}$ multilayers (Supplementary Figs. 22 and 23). As a result, the electrochemical performance of the (GOx/PEI)$_{30}$/20-MCF electrode was lower (i.e. a low anodic areal current density, a high internal resistance and a low ion-diffusion rate) than the (GOx/TREN)$_{30}$/20-MCF (i.e., 30-GOx/20-MCF) electrode (Fig. 4b, c, Supplementary Figs. 24 and 25), further confirming the role of the small-molecule linker. This conformal coating of the thin GOx layer on the conductive MCF facilitates charge transfer and thereby maximizes the catalytic reaction of GOx. Additionally, we expect that the anodic performance can be further controlled and enhanced with increasing the electrical conductivity of MCF for facile electrical communication and the surface area of MCF for efficient enzyme loading (by a further increase (i.e., $n > 20$) of bilayer number of (TOA-Au NP/TREN)$_n$ multilayers onto cotton fibers).

**Performance of hybrid metallic cotton fiber-biofuel cells**. To demonstrate a working MCF-BFC, the 30-GOx/20-MCF anode was connected to the 20-MCF cathode and the power density was

evaluated by measuring the current flow through a fixed external resistance in the range of 1 kΩ−10 MΩ without membrane (Supplementary Fig. 26a). The resulting MCF-BFC delivered the maximum areal power density of approximately 1.4 mW cm$^{-2}$ at the potential of +0.37 V in the electrolyte containing 300 mmol L$^{-1}$ glucose under ambient condition (Supplementary Fig. 26b). Although the power density at the lower concentration of 10 mmol L$^{-1}$ was decreased to ~0.9 mW cm$^{-2}$, this performance was still sufficiently high to support various in vivo applications[59], such as self-powering glucose sensors[60] or implantable power supplies for cardiac pacemaker[61], neurostimulator[62], and drug pumps[63]. The MCF-BFC also exhibited high operational stability, maintaining ~71% (~0.6 mW cm$^{-2}$) of its initial power density and voltage retention of ~83% after continuous operations for 20 days (~62% of the initial power density and voltage retention of ~79% after 35 days) (Supplementary Fig. 26c, d).

**Power output enhancement of hybrid metallic cotton fiber-biofuel cells**. The power output of MCF-BFCs can be further improved by enhancing the active surface area and electrical conductivity of the cathode electrode. We systematically controlled the number of bilayers ($n$) of (TOA-Au NP/TREN)$_n$ from 20 to 120 in the cathode and investigated its effect on the power performance of MCF-BFCs (Fig. 5a). Through this process, the electrical conductivity of $n$-MCFs increased to approximately

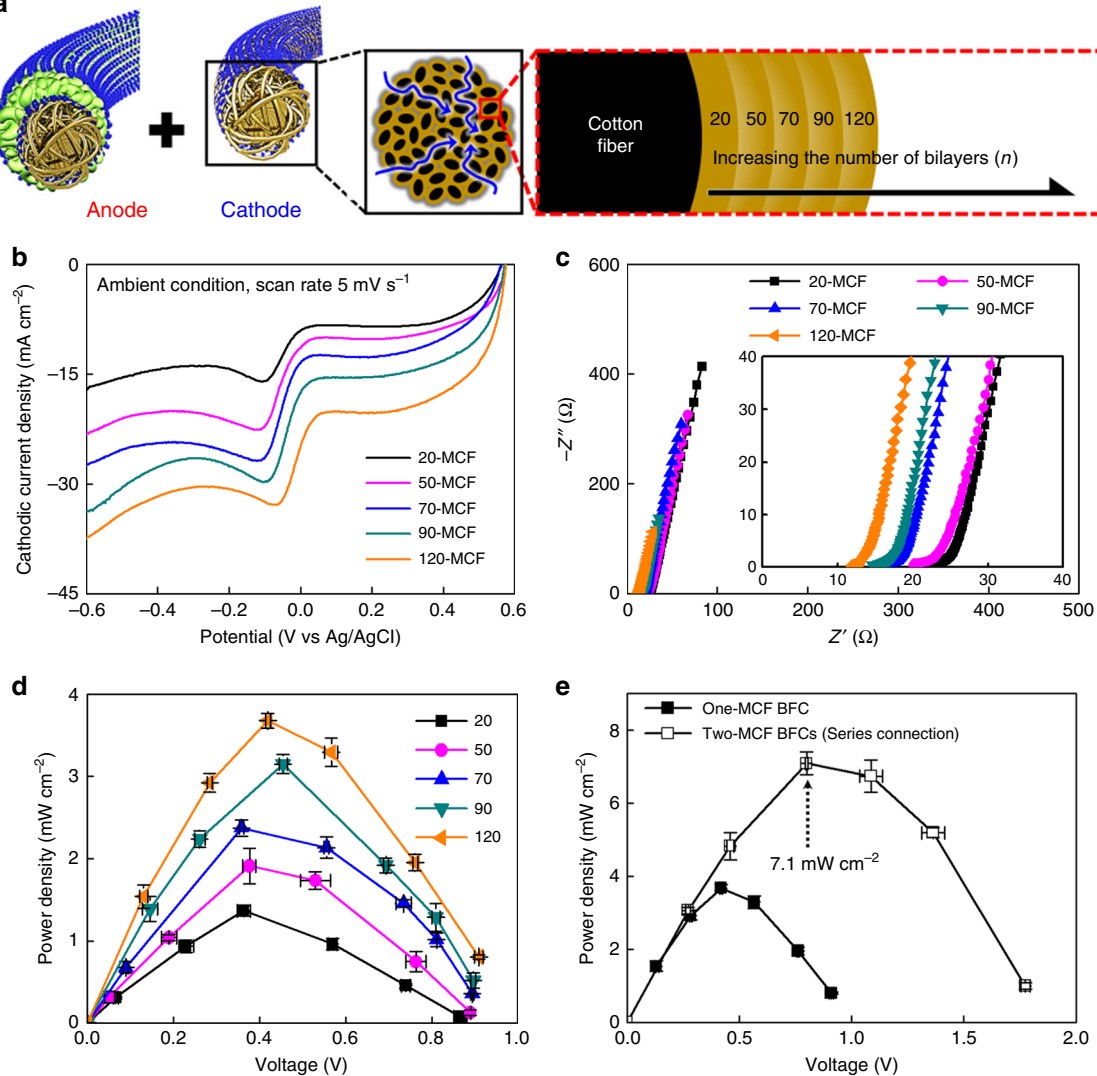

**Fig. 5** Enhancement of the power output of a metallic cotton fiber-based biofuel cell. **a** Complete metallic cotton fiber-based biofuel cells (MCF-BFCs) with a constant anode (30-bilayered glucose oxidase (GOx)/tris-(2-aminoethyl)amine (TREN)) multilayer-coated MCF (i.e. 30-GOx/20-MCF) and cathodes with a controlled number of bilayers n (n-bilayered metallic cotton fiber, n-MCF). **b** Cathodic current curves for n-MCF cathodes as a function of n. **c** Nyquist plots of the MCF cathodes depending on n. In this case, the equivalent series resistance (ESR) values for $n = 20, 50, 70, 90$, and 120 were approximately 24, 22, 17, 16, and 13 Ω, respectively. **d** Power output of the MCF-BFCs (an n-MCF cathode and a 30-bilayered GOx/TREN multilayer-coated 20-bilayered MCF anode) with external resistors (1 kΩ–10 MΩ) as a function of n-MCF. **e** Power outputs of one and two MCF-BFCs composed of the 120-MCF cathode and the 30-GOx/20-MCF anode. In these cases, the two MCF-BFCs were connected with each other in series, and its power output was also measured with current output through operating a cell potential by an external resistor. The error bars (**d**, **e**) show the standard deviation from the mean value of power densities for 3–5 independent experiments

$2.1 \times 10^4$ S cm$^{-1}$ (Au NP/TREN loading density of ~51.7 mg cm$^{-2}$) at $n = 120$ (Supplementary Fig. 27). At a potential of −0.6 V, the cathodic current density gradually increased from −17 mA cm$^{-2}$ for 20-MCF to −38 mA cm$^{-2}$ for 120-MCF (Fig. 5b and Supplementary Fig. 28). Additionally, the ORR peaks progressively shifted to positive potentials with an increasing number of bilayers, which can be attributed to the increased electrical conductivity and decreased overpotential for the ORR[64]. This was further confirmed by the decreased ESR values of n-MCFs from approximately 24 to 13 Ω as n was increased (Fig. 5c). The assembled cathodes with different numbers of bilayers were connected to a constant anode (i.e., 30-GOx/20-MCF) with an external resistor for the preparation of complete BFCs. The maximum areal power densities were found to increase gradually

with the number of bilayers of the cathode (Fig. 5d and Supplementary Fig. 29). As a result, our MCF-BFC composed of a 120-MCF cathode and a 30-GOx/20-MCF anode (i.e. 120-MCF-BFC) exhibited the maximum areal power density of 3.7 mW cm$^{-2}$ in 300 mmol L$^{-1}$ glucose under ambient condition applying a fixed external resistance, considerably outperforming conventional BFCs (including MET-BFCs) reported to date (Supplementary Table 1). This high-power output (3.7 mW cm$^{-2}$) obtained from our membrane-free and mediator-free MCF-BFC is in stark contrast with those measured from other mediator-free enzymatic fuel cells (i.e. in the case of being measured by an external resistor, the power output of mediator-free enzymatic fuel cells is generally less than ~1 mW cm$^{-2}$). For example, the power output shown in our system is nearly three times higher than that of the previous

DET-based compressed CNT electrode[5] (1.25 mW cm$^{-2}$, measured by polarization tests, i.e. CV measurements, without an external resistor), and even higher than that of the reported osmium complex-based bi-scrolled carbon nanotube yarn MET-BFC[7] (2.18 mW cm$^{-2}$). Consequently, we demonstrated that our highly porous metallic cotton fiber-based BFC (MCF-BFC) fabricated by small-molecule linker-induced LbL assembly of the active catalysts can significantly realize the DET rate between the enzymes and the conductive supports. More significantly, the maximum power density of an MCF-BFC can be further increased by simply increasing the bilayers of Au NP layers in the cathode and/or in the anode. Moreover, these controlled electrode–electrode and/or BFC–BFC connections can further increase the overall power density of the device. For example, if two 120-MCF-BFCs were connected in series, the maximum power output measured was ~7.1 mW cm$^{-2}$, which is an approximately twofold increase in the hybrid BFC performance (Fig. 5e).

## Discussion

We have demonstrated ultrahigh-power hybrid DET-BFCs by using small-molecule-linker-derived metal NPs and enzyme multilayers that not only stimulate electrical communication between the electrochemically active components but also maximize the electrocatalytic activity on highly porous cotton fibers. In addition, we demonstrated that the power performance of the MCF-BFCs could be further enhanced by increasing the electrical conductivity and the electrocatalytic surface area of the MCF through increasing the number of TOA-Au NP/TREN bilayers. Consequently, the areal power density of 3.7 mW cm$^{-2}$ obtained from the 120-MCF-BFC using a fixed external resistance exhibited the best-reported performance of the BFCs. This result emphasizes the ability of the small-molecule linker-induced LbL assembly to fine-tune the interfacial adsorption onto the highly porous cotton fibers and the structure between the metal NPs and the enzymes, thereby enhancing their electrochemical performance. We believe that our approach can be widely applied to electrode designs for various electrochemical energy storage and conversion devices.

## Methods

**Materials**. Cotton fibers with a diameter of 200 μm were purchased from Igonji (South Korea), and glucose oxidase (GOx) from *Aspergillus niger* (180 U mg$^{-1}$) was purchased from Amano Enzyme Inc. (Japan). Tetraoctylammonium bromide (TOA), gold (III) chloride trihydrate (HAuCl$_4$·3H$_2$O), sodium borohydride (NaBH$_4$), tris-(2-aminoethyl)amine (TREN) ($M_w$ ~146), and polyethylenimine, branched (PEI, $M_w$ ~800 and 25,000) were purchased from Sigma-Aldrich.

**Synthesis of tetraoctylammonium bromide-stabilized gold nanoparticles**. Tetraoctylammonium bromide-stabilized Au NPs (TOA-Au NPs) with a diameter of 8 nm were synthesized using a two-phase method in toluene[43]. In brief, HAuCl$_4$·3H$_2$O in water (30 mmol L$^{-1}$, 30 mL) and TOA in toluene (20 mmol L$^{-1}$, 80 mL) were mixed with vigorous stirring, and then NaBH$_4$ in water (0.4 mol L$^{-1}$, 25 mL) was added to the two-phase mixture. After sufficient stirring (approximately 3 h), the aqueous solution was separated from the mixture, and then the remaining toluene solution was repeatedly washed using dilute aqueous H$_2$SO$_4$ (0.1 mol L$^{-1}$), NaOH (0.1 mol L$^{-1}$), and deionized water. After several washing steps and the removal of the aqueous solution, TOA-Au NPs dispersed in toluene were obtained.

**Fabrication of metallic cotton fiber electrodes**. Cotton fibers composed of hydroxy (OH) group-functionalized cellulose fibers were dipped in ethanolic solutions of amine (NH$_2$)-functionalized PEI ($M_w$ ~800; 1 mg mL$^{-1}$) for 40 min. The resulting PEI-coated fibers, which were prepared through a hydrogen-bonding interaction between the NH$_2$ of PEI and the OH groups of the cotton fibers, were then immersed in a TOA-Au NP solution (10 mg mL$^{-1}$) for 40 min, and then washed with toluene to remove the weakly adsorbed TOA-Au NPs. In this case, the bulky TOA ligands bound to the surface of the Au NPs were almost completely replaced by NH$_2$ groups of the PEI-coated fibers due to the high affinity between the Au surface and NH$_2$ groups. Next, the formed TOA-Au NP-coated fibers were

immersed in a solution of TREN in ethanol (1 mg mL$^{-1}$) for 40 min. This was repeated, and the fibers were washed in ethanol to remove the excess TREN molecules, resulting in fibers coated with one bilayer (i.e. (TOA-Au NP/TREN)$_1$-cotton fiber). The adsorption mechanism between the TOA-Au NPs and the NH$_2$-functionalized TREN is the same as that between the TOA-Au NPs and PEI. These deposition and washing steps for the preparation of the MCFs were continuously repeated until the desired number of bilayers was obtained.

**Fabrication of anode electrodes**. The fabricated MCF electrode was immersed in a solution of GOx (5 mg mL$^{-1}$ in PBS) for 10 min, and then washed with PBS (pH 7.4) to remove weakly adsorbed GOx. During these deposition and washing steps, negatively charged GOx molecules in the pH 7.4 PBS aqueous solution were electrostatically adsorbed onto the positively charged MCF. It should be noted that the neutral NH$_2$ groups of the outermost TREN layer of the MCF prepared in organic phases can be easily protonated to positively charged ammonium (NH$_3^+$) groups in pH 7.4 PBS aqueous solution because the p$K_a$ (pH with a degree of ionization of 50%) of ammonium groups is approximately 10. After the electrostatic deposition of GOx, a solution of positively charged TREN (1 mg mL$^{-1}$) was deposited on the anionic GOx-coated MCFs for 10 min, which were then washed with PBS. These processes were repeated until an anode electrode (i.e. (GOx/TREN)$_m$-coated MCF) with the desired number of bilayers ($m$) was obtained. To prepare the polymer linker-based fiber anode electrode, a solution of cationic PEI ($M_w$ ~25,000; 1 mg mL$^{-1}$) was used instead of TREN. In this case, all the LbL assembly conditions, except the inclusion of polymer, were exactly the same as those used for the (GOx/TREN)$_m$ multilayers.

**Characterization and analysis of metallic cotton fiber-based electrodes**. The surface and cross-sectional morphologies of the MCF-based BFC electrodes were characterized using FE-SEM (Hitachi S4700, Japan) and AFM (XE-100, Park Systems, South Korea). The adsorption behavior and mechanism of formation of the (TOA-Au NP/TREN)$_n$ multilayers were investigated using FTIR spectroscopy in specular mode (Cary 600 spectrometer, Agilent Technologies, USA). The growth and loading amount of the LbL-assembled multilayers were examined using UV−Vis spectroscopy (Lambda 35, Perkin Elmer) on quartz glass, and a QCM (QCM 200, SRS, USA), respectively. In the case of QCM measurement, the mass change (Δ$M$) per layer was calculated from the QCM frequency change (Δ$F$) using the Sauerbrey equation[65],

$$\Delta F(\text{Hz}) = -\frac{2F_0^2}{A\sqrt{\rho_q\mu_q}} \cdot \Delta M,$$

where $F_0$, $A$, $\rho_q$, and $\mu_q$ represent the resonant frequency (~5 MHz), the active area (cm$^2$), the density (2.65 g cm$^{-3}$), and the shear modulus (2.95×10$^{11}$ g cm$^{-1}$ s$^{-2}$) of the QCM electrode, respectively, and this expression can be simplified to Δ$F$ (Hz) = −56.6 × Δ$M_A$ ($M_A$ indicates mass per unit area).

**Electrochemical measurements**. In a three-electrode system composed of an MCF working electrode, Ag/AgCl reference electrode, and a Pt counter electrode, the top end of the working electrode was firmly connected to a copper wire using commercial silver paste (ELCOAT P-100, CANS, South Korea). After the silver paste was dry, it was additionally insulated with an epoxy adhesive. Cyclic voltammograms of an MCF electrode (diameter 200 μm, length 5.0-mm, active external surface area 3.14 mm$^2$) were measured over the potential range −0.6 to +0.6 V using an electrochemical analyzer (Ivium-n-Stat, Ivium Technologies, Netherlands). All measurements were carried out in an electrochemical cell in batch mode, 50 mL PBS buffer (20 mmol L$^{-1}$ phosphate and 0.14 mol L$^{-1}$ NaCl, pH 7.4) at 36.5 °C without stirring. Interelectrode spacing between anode and cathode electrode was ~1 cm, the membrane between two electrodes was not used in this study. In addition, one end of each BFC electrode was fixed by epoxy resin on the glass substrate for avoiding electrical shortage while measuring the power density with the continuous operation. The anodic and cathodic current densities were determined by CV at a constant voltage of +0.6 V for the anode and −0.6 V for the cathode, showing a maximum current density. The normalized current density was respectively obtained by subtracting the current density level in glucose-free PBS at +0.6 V for the anode, and subtracting the current density level in an oxygen-free PBS at −0.6 V for the cathode. The BFC power densities were determined by measuring the current output applying a fixed external resistance (in the range of 1 kΩ~10 MΩ) to control cell potential[3,5,66]. The experimental setup for measuring the power density of the complete hybrid MCF-BFC was shown in Supplementary Figure 26a. EIS measurements for the $n$-MCF electrodes and the $m$-GOx/$n$-MCF electrodes were performed over the frequency range 0.2 Hz−100 kHz with a perturbation amplitude of 0.01 V. The impedance spectra (Nyquist plots) of the real ($Z'$) and imaginary parts ($Z''$) obtained from the electrochemical BFC cell were processed using the ZView software (ver. 2.8d, Scribner Associates Inc., USA).

## Data availability
The authors declare that the data supporting the findings of this study are available from the corresponding author upon request.

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

## Acknowledgements

This work was supported by a National Research Foundation (NRF) grant funded by the Ministry of Science, ICT & Future Planning (MSIP) (2018R1A2A1A05019452; 2016M3A7B4910619) and the Basic Science Research Program through the National Research Foundation of Korea (NRF) funded by the Ministry of Education (NRF-2017R1A6A3A04003192).

## Author contributions

C.H.K., Y.K., S.W.L. and J.C. conceived the idea and designed the experiments. C.H.K., Y.K., D.S., M.K. and W.K.B. performed all the experiments. C.H.K., Y.K., J.P, S.W.L. and J.C. wrote and revised the manuscript. All authors discussed the results and commented on the manuscript.

## Additional information

**Competing interests:** The authors declare no competing interests.

