## [Peer Review File · Nature Communications]

Reviewers' comments:

Reviewer #1 (Remarks to the Author):

This paper reports the metalized cotton fiber (MCF) electrodes and application to biofuel cells. The MCFs prepared by LbL coating of AuNP showed activity for O₂ reduction (cathode reaction). The further coating of the MCF with multilayer of enzyme (GOD) exhibited catalytic oxidation of glucose (anode reaction).

This work is composed of following 4 topics.

(1) Development of conductive, porous MCF electrodes, (2) O₂ reduction at the MCF electrode, (3) Oxidation of glucose at the enzyme modified MCF electrodes without mediator, and (4) BFC shows superior performance.

Regarding to the topic 1: The LbL techniques have been widely used for functional coating. It would be novel and interesting that this study used small linker molecule TREN to get higher, stable conductivity.

Topic 2: O₂ reduction at AuNP modified nanostructured electrodes have been already reported so far (ref. 46, 47 and so on).

Topic 3: It is seriously strange that the GOD multilayer serves as electrocatalyst for glucose oxidation even without any electron mediator. The GOD multilayer is electrically insulating. Author should explain the reaction mechanism, and the reason why the electrode activity was amplified by the coating of GOD multilayer. Importantly, authors reported in their previous paper (J. Power Sources, 286, 197, 2015) that GOD multilayer more than 2 layer can't work.

Topic 4: The output power obtained in this work, 3.7 mWcm⁻², is not very special. There are papers reporting the power in the range of mW/cm². ECS Trans. 2009 16(38): 9-15

In this work, I can't find novelty and impact enough for publication in Nature Commun. I recommend authors to sincerely explain the mechanism of glucose oxidation at GOD multilayer-coated electrode in their revised version, and resubmit to more specialized journal.

Reviewer #2 (Remarks to the Author):

The authors report here the development of a hybrid biofuel cells that employes a glucose oxidase anode, made of metallic cotton fibres, and a cathode made of the same electrode.

Overall the manuscript is interesting and very well presented. Particularly appreciated are the characterisation made on the electrodes, both material characterisation and electrochemical characterisation, which provides a very comprehensive and systematic study. As such, this Reviewer suggests publications, after major revision.

One aspect the authors should focus on and enhance is the part related to the fuel cell operation, as most of the manuscript focuses on the development and characterisation of the electrodes only. the results should be better discussed with reference on existing literature and particularly mediator-free enzymatic fuel cells.

It is noted that the graphs related to the electrochemical characterisation of the electrodes and of the fuel cells, have no error bars. Have the authors done any replicates of the experiments? what is the reproducibility of the results?

Can the enzymatic loading be estimated? considering that the immobilisation relies on an electrostatic interaction, how reproducible electrodes are and how stable? Have the authors experienced any leaching over prolonged operation?

Was the fuel cell performance assessed only via polarisation tests or was the voltage versus time recorded for long period of times?

Does Figure 22c refer to peak power values obtained from polarisation tests or does it refer to

continuous operation in buffer? For practical application tests on performance under continuous operation is key.

Response to Reviewers' Comments

Title: **High-Power Hybrid Biofuel Cells Using Layer-by-Layer Assembled Glucose Oxidase-Coated Metallic Cotton Fibers**

We appreciate the comments and suggestions raised by the reviewers. Our responses are listed below, and the main text and supplemental materials file have been modified according to the referees' comments.

REVIEWERS' COMMENTS:

Reviewer #1

Overall comments to the Author: *This paper reports the metalized cotton fiber (MCF) electrodes and application to biofuel cells. The MCFs prepared by LbL coating of AuNP showed for O₂ reduction (cathode reaction). The further coating of the MCF with multilayer of enzyme (GOD) exhibited catalytic oxidation of glucose (anode reaction).*

This work is composed of following 4 topics.

(1) Development of conductive, porous MCF electrodes, (2) O₂ reduction at the MCF electrode, (3) Oxidation of glucose at the enzyme modified MCF electrodes without mediator, and (4) BFC shows superior performance.

Regarding to the topic 1: The LbL technique have been widely used for functional coating. It would be novel and interesting that this study used small linker molecule TREN to get higher, stable conductivity.

Topic 2: O₂ reduction at AuNP modified nanostructured electrodes have been already reported so far (ref. 46, 47 and so on).

Topic 3: It is seriously strange that the GOD multilayer serves as electrocatalyst for glucose oxidation even without any electron mediator. The GOD multilayer is electrically insulating. Author should explain the reaction mechanism, and the reason why the electrode activity was amplified by the coating of GOD multilayer. Importantly, authors reported in their previous paper (J. Power Sources, 286, 197, 2015) that GOD multilayer more than 2 layer can't work.

Topic 4: The output power obtained in this work, 3.7 mW/cm², is not very special. There are papers reporting the power in the range of mW/cm². ECS Trans. 2009 16(38): 9-15

In this work, I can't find novelty and impact enough for publication in Nature Commun. I recommend authors to sincerely explain the mechanism of glucose oxidation at GOD multilayer-coated electrode in their revised version, and resubmit to more specialized journal.

Comment 1. *O₂ reduction at AuNP modified nanostructure electrodes have been already reported so far (ref. 46, 47 and so on).*

Response 1: We appreciate reviewer #1's comment. As already mentioned in our original manuscript, other research groups [*J. Phys. Chem. C* **112**, 13886–13862 (2008); *ACS Appl. Mater. Interfaces* **8**, 20635–20641 (2016)] reported that the Au nanoparticles (NPs) formed onto carbon materials could be used as oxygen reduction reaction (ORR) catalysts rather than electrical conductors. **However, we have demonstrated for the first time that the layer-by-layer assembled Au NPs can be effectively used as bulk metal-like electron conductors as well as ORR catalysts for biofuel cell system, and furthermore, their performance can be easily controlled and enhanced through the controlled incorporation of Au NPs within cellulose cotton fibers.** That is, the originality of our research lies in the fact that our approach can provide both the metallic conductivity and the catalytic property (*i.e.*, ORR at cathode) to the highly porous cellulose cotton fibers through a simple LbL assembly method. In terms of the performance of the fuel cell systems, the ORR at the cathode is a critical factor in determining the fuel cell output due to its complex and limited electron-transfer kinetics features. At the same time, the mass loading of the ORR catalyst is also significantly related to the cathode performance. In the present study, we showed that the ORR peak current density could be gradually increased with increasing the bilayer number (n) (or mass of the Au NPs) of the (TOA-Au NP/TREN) $_n$ multilayers onto the cellulose fiber substrate (see **Figure 5b** in the revised manuscript), which was attributed to the improved electron transfer kinetics confirmed by the electrochemical impedance measurement (see **Figure 5c** in the revised manuscript). More importantly, the obtained results are based on the dense and uniform deposition of Au NPs on the cellulose fibers by small molecule linker-based LbL assembly in organic media accompanying the ligand exchange reaction between bulky hydrophobic ligand (*i.e.*, TOA) and amine moiety of TREN, which cannot be easily achieved with the conventional electrostatic LbL assembly in aqueous media or the physical mixing method. Thus, we emphasize that, despite the use of the same materials, our unique approach for designing the electrode is clearly distinguished from the previous reports where the ORR current range is less than $\sim 1 \text{ mA cm}^{-2}$ (including ref. 46, 47 and so on).

On the revised manuscript:

Figure 5. **b**, Cathodic current curves for n -MCF cathodes as a function of n . **c**, Nyquist plots of the MCF cathodes depending on n . In this case, the ESR values for $n = 20, 50, 70, 90,$ and 120 were approximately $24, 22, 17, 16,$ and 13Ω , respectively.

Comment 2. Importantly, author reported in their previous paper (*J. Power Source*, 286, 197, 2015) that GOD multilayer more than 2 layer can't work.

Response 2: We believe that the reviewer confused the work of another group with our group's previous works. **We should mention that our previous research has nothing to do with the paper reported by Prof. Yongchai Kwon and colleagues** [*J. Power Source*, 286, 197–203 (2015)]. However, for resolving the concerns raised by reviewer #1, we explain the difference between Prof. Yongchai Kwon's results and our current study from the viewpoint of device structure and interfacial chemistry.

In the paper reported by Prof. Yongchai Kwon, they described that the $(\text{GOx/PEI})_n$ multilayers were LbL-assembled onto the 20 nm-thick bare multiwall carbon nanotube (MWCNT) without any chemical treatment, and additionally, these $(\text{GOx/PEI})_n$ multilayer-coated MWCNTs and Nafion were drop-casted onto glass carbon electrodes (GCEs) for electrochemical measurements. That is, **the biofuel cells reported by Prof. Kwon and colleagues were based on flat film electrode** [reference to redacted figure] **instead of implantable thread or fiber-type electrodes shown in our study** (see Figure R1). Furthermore, they did not provide any information about the total thickness of the solution-casted $(\text{GOx/PEI})_2$ multilayer-coated MWCNT films although they reported that the thickness of $(\text{GOx/PEI})_2$ multilayers was ~ 2.5 nm.

[redacted figure]

Figure R1. Cross-sectional FE-SEM images of metallic cotton fiber (MCF)-based electrode in our study.

Additionally, from the perspective of the LbL assembly, it is unreasonable that the polyelectrolyte multilayers composed of bulky anionic GOx and cationic PEI [molecular weight (M_w) of PEI used in their study $\sim 750,000$] are uniformly coated onto 20 nm-thick bare MWCNTs without any chemical treatment. Therefore, we have much difficulty in accepting their results that the best power performance of biofuel cell was obtained from $(\text{GOx/PEI})_2$ -MWCNT-based electrodes, and additionally, the thickness of $(\text{GOx/PEI})_2$ multilayer was approximately 2.5 nm. The detailed explanations about our concerns are as follows:

When the LbL assembly of the $(\text{GOx/PEI})_n$ multilayers was performed onto uncharged bare MWCNT, many processing steps (*i.e.*, the deposition of polymer, the sedimentation of polymer-coated MWCNTs, the removal process of supernatant solution, the washing processes for removal of weakly adsorbed polymers, and then the ultrasonication for dispersion of polymer-coated MWCNTs) were repeatedly required for the coating of polymeric multilayers such as $(\text{GOx/PEI})_n$ multilayers. However, these 20 nm-thick MWCNTs are seriously agglomerated and sedimented even after the first deposition of bulky PEI layer with high M_w because of polymeric bridging between neighboring MWCNTs. It should be noted that in the case of small colloids with a diameter below 100 nm (*i.e.*, anionic polystyrene or silica colloids), electrostatic LbL assembly using polyelectrolyte linkers is very difficult. Additionally, in the case of using the uncharged bare MWCNTs instead of anionic MWCNT with the COO^- functional groups, these phenomena are generally accelerated during repeated LbL

assembly process [these aggregation phenomena are well explained in the previous paper reported by Frank Caruso, *Macromolecules*, **35**, 9780–9787 (2002)]. Therefore, it is reasonable to analogize that the periodic zeta-potential changes obtained from the alternating deposition of PEI and GOx onto MWCNTs in their paper are mainly induced by the presence of excess PEI and GOx onto MWCNTs.

To clarify this issue, we have prepared the GOx-based multilayer using PEI [*i.e.*, (GOx/PEI)₃₀] with M_w of $\sim 750,000$ used in Yongchai Kwon's group [*J. Power Source*, **286**, 197, (2015)] under the same preparation condition, and compared to the (GOx/TREN)₃₀ multilayers in our system (see **Figure R2**). As a result, the (PEI/GOx)₃₀ multilayers displayed the film thickness of $\sim 318 \pm 7$ nm, while the (GOx/TREN)₃₀ multilayers showed the thickness of $\sim 13 \pm 5$ nm, which were measured by cross-sectional FE-SEM images. In other words, the average thickness of the enzyme layer (*i.e.*, GOx/each linker) deposited per 1 bilayer can be estimated to be significantly different values of ~ 10.6 nm for GOx/PEI and ~ 0.43 nm for GOx/TREN layer. Although we used the Au-coated Si wafer as a substrate to deposit the enzyme layer, the obtained results clearly indicate that the polymeric aggregation phenomenon can be accelerated when using the bulky PEI linker instead of the small-molecule TREN.

Figure R2. Cross-sectional FE-SEM images for film thickness of **a**, (GOx/PEI)₃₀ and **b**, (GOx/TREN)₃₀ multilayers deposited onto the Au-coated Si wafer.

Furthermore, the respective PEI layers sandwiched between adjacent GOx layers can seriously limit the electron transport and tunneling from GOx to electrode despite the use of low bilayer number (*i.e.*, 2 or 3 bilayered GOx/PEI) because of their bulky size (M_w of PEI $\sim 750,000$) and insulating characteristic.

On the other hand, we highlight that the bilayer-controlled GOx/TREN multilayers shown in our study are very similar to the thickness-controlled single GOx layer due to the use of TREN ($M_w \sim 146$) as a small organic linker. This possibility can be confirmed by our experimental results on the linker-dependent current density performance for (GOx/linker)_{*m*}, and conductive Au-based layer (Au NP/linker)_{*n*}, (**Figure 4b, c** and **Supplementary Figure 20** in the revised manuscript and Supplementary Information). In this case, PEI-based electrodes showed poor electrochemical performance compared to TREN-based electrodes in both the anode and cathode. These results strongly suggest that the use of bulky polymer linker such as PEI for the preparation of GOx-based multilayers makes a fatal effect on the electron transport between the GOx multilayer and the metal electrode.

On the revised manuscript:

Figure 4. b, Anodic current density curves of 30-GOx/20-MCF electrodes prepared from (GOx/PEI)₃₀ and (GOx/TREN)₃₀ in the absence and presence of glucose (300 mmol l⁻¹). **c**, Nyquist plots for the 30-GOx/20-MCF anodes prepared using PEI and TREN linkers. The ESR values of the PEI- and TREN-based 30-GOx/20-MCF anodes were 62 and 40 Ω, respectively, in PBS containing 300 mmol l⁻¹ glucose.

On the revised Supplementary Information:

Supplementary Figure 20. a, Anodic current density of (GOx/TREN)₃₀(TOA-Au NP/PEI)₂₀ electrode as a function of glucose concentration from 0 to 300 mmol l⁻¹. **b**, Comparison of the normalized current densities of (GOx/TREN)₃₀(TOA-Au NP/PEI)₂₀ and (GOx/TREN)₃₀(TOA-Au NP/TREN)₂₀ anode electrodes with different glucose concentration recorded at +0.6 V.

Comment 3. *It is seriously strange that the GOD multilayer serves as electrocatalyst for glucose oxidation even without any electron mediator. The GOD multilayer is electrically insulating. Author should explain the reaction mechanism, and the reason why the electrode activity was amplified by the coating of GOD multilayer.*

Response 3: We appreciate reviewer #1's comment. As the reviewer pointed out, the thickness of GOx/TREN multilayers is a critical factor for electron transfer and catalytic performance because the electron communication between GOx and electrode is based on electron tunneling mechanism. Particularly, in the case of LbL assembly using bulky GOx, the multilayer thickness can exceed the tunneling thickness. Therefore, controlling the adsorption amount and thickness of GOx is most important with the small organic linkers (*i.e.*, TREN) for efficient electron transfer.

To this end, we delicately controlled the thickness (or loading amount) of GOx/TREN multilayers using solution concentration-controlled electrostatic LbL assembly of GOx (5 mg ml^{-1}) and TREN (1 mg ml^{-1}) solution at human-body compatible pH 7.4. First, it should be noted that the pK_a (pH with a degree of ionization of 50 %) of carboxylic acid groups within GOx is approximately 4.5, and therefore GOx at pH 7.4 is highly negatively charged. On the other hand, in the case of TREN with $pK_a \sim 10$, TREN at pH 7.4 is highly positively charged. **These highly charged GOx and TREN generate the extremely low loading amount per layer because of long-range electrostatic repulsion between the same charged GOx as well as the same charged TREN.** In the case of measuring the film thickness using Au-coated Si substrates similar to Au NP-coated cotton fibers (*i.e.*, metallic cotton fiber, MCF), **the thicknesses of (GOx/TREN)₂₀ and (GOx/TREN)₃₀ multilayers were measured to be $\sim 9 \pm 3 \text{ nm}$ and $\sim 13 \pm 5 \text{ nm}$, respectively despite deposition of 20 or 30 bilayers** (see Figure R3). **Considering the dimension of GOx ($6.0 \times 5.2 \times 7.7 \text{ nm}^3$) [Sensors, 8, 5637–5648 (2008).], these thicknesses of the multilayers indicate that GOx enzymes are adsorbed on the substrates as a single layer for 20 bilayer-film and as a bi-layer of 30 bilayer-film.** Thus, the progressive increase of the anodic current up to 20 bilayers can be attributed to the gradual adsorption of GOx enzymes as the single layer form on the substrate (Figure 3c). On the other hand, the minor increase of the anodic current from 20 to 30 bilayers may be due to the adsorption of the GOx enzymes as the double-layer form that can interfere the electron communication between the GOx and the Au NPs (Figure 3c).

Figure R3. Cross-sectional FE-SEM images for film thickness of **a**, (GOx/TREN)₂₀ and **b**, (GOx/TREN)₃₀ multilayers deposited onto the Au-coated Si wafer.

Figure 3. c, Anodic current density curves of the $(\text{GOx}/\text{TREN})_m/20\text{-MCF}$ electrode. Inset: the change in the normalized anodic current density levels at +0.6 V; here, the normalized level is defined as the current density level for which the anodic current density level of 20-MCF ($m=0$, black curve) is subtracted from the anodic current density level of m -GOx/20-MCF at the specific potential of +0.6 V. The error bars show the standard deviation from the mean value of current densities for three to five independent experiments.

It should also be noted that the thickness (or loading amount) of GOx/TREN multilayers onto the outer surface of MCFs can be notably different from the multilayer thickness onto the interior of MCFs. As already mentioned in our manuscript, it is not easy for bulky GOx to be deeply incorporated into the interior of highly porous MCFs due to the blocking-up phenomena. Although we could not quantitatively investigate the thickness of GOx/TREN multilayers coated onto the inside cotton fibers, it is reasonable to analogize that the thicknesses of $(\text{GOx}/\text{TREN})_{20}$ and $(\text{GOx}/\text{TREN})_{30}$ multilayers deposited onto the interior of MCFs may be thinner (presumably, corresponding to thickness of 2-3 nm for electron tunneling) than the outermost surface of metallic cotton fibers, and additionally facilitate electron tunneling from GOx within porous metal frame to metallic electrodes.

Furthermore, we highlight that the equivalent series resistance (ESR) value of highly porous MCFs is much lower than that of $(\text{GOx}/\text{TREN})_m$ multilayer-coated nonporous and flat substrates with the same bilayer number (n) (i.e., Au NP-coated Si substrates). For example, in the case of the $(\text{GOx}/\text{TREN})_{20}/\text{MCF}$ showing almost saturated anodic current density and the $(\text{GOx}/\text{TREN})_{20}/\text{nonporous}$ substrate, their ESR values at 1 kHz were measured to be ~ 38.6 and ~ 62.6 Ω , respectively (see **Figure R4**), and resultantly the anodic current densities of $(\text{GOx}/\text{TREN})_{20}$ multilayer-coated MCF electrode outperformed those of $(\text{GOx}/\text{TREN})_{20}$ multilayer-coated nonporous electrodes (see **Supplementary Figure 18** in the revised Supplementary Information).

Figure R4. Nyquist plots of the (GOx/TREN)₂₀ multilayers coated onto MCF (red solid circle) and nonporous plate substrate (i.e., Au-coated Si wafer substrate, blue solid square). In this case, ESR values of each electrode were measured to be ~38.6 Ω for 20-GOx/MCF and ~62.6 Ω for 20-GOx/Au-coated Si wafer electrodes, respectively.

Supplementary Figure 18. Electrochemical performance of flat substrate-based anodes. a, Anodic current density curves. b, The normalized current density levels of cotton substrate- and Si wafer-based anodes measured at +0.6 V as a function of m . All the measurements were performed at a scan rate of 5 mV s^{-1} in a PBS containing 300 mmol l^{-1} glucose under ambient conditions. Here, the normalized anodic current density performance is $\sim 0.01 \text{ mA cm}^{-2}$ for the Si wafer substrate at 300 mmol l^{-1} glucose, compared to the value of the cotton substrate, $\sim 33.6 \text{ mA cm}^{-2}$ ($m = 20$).

We also observed that the electron transfer kinetics at the interface of the electrode are significantly influenced by a kind of outermost layer of GOx/TREN multilayers (see **Figure R5**). More specifically, OH-functionalized glucose has a high affinity (mainly, hydrogen-bonding interaction) with amine-functionalized TREN than negatively charged GOx (due to the presence of carboxylate ion (COO^-) groups of GOx). In line with this high affinity, glucose could be easily diffused into outermost TREN-coated multilayer films, and therefore the outermost TREN-coated electrode exhibited much lower electron transfer resistance than outermost GOx-coated electrodes in spite of the increase of layer number. These similar phenomena were also demonstrated by Willner group [*Langmuir*, **17**, 1110-1118 (2001)]. They have shown that the electron transfer resistance at the interface between the electrode and the electrolyte can be controlled by the electrostatic attraction or repulsion between the charged outermost layer and the redox label in the electrolyte [i.e., $\text{FeI}(\text{CN})_6^{3-/4-}$]. Considering that all the biofuel cells shown in our study are based on the outermost TREN-coated electrodes as well as highly porous electrodes, the electron transfer resistances of metallic cotton fiber-based electrodes can

^{a)}The interfacial electron transfer resistance between the charged interface of the electrode and the electrolyte.

be significantly lowered compared to those of other nonporous electrodes.

Figure R5. Outermost layer-dependent interfacial electron transfer kinetics (tested on the Au-coated Si wafer substrate). **a**, Nyquist plots of $(\text{GOx}/\text{TREN})_m$ multilayers as a function of bilayer number (m). **b**, Representative schematics of $(\text{GOx}/\text{TREN})_m$ multilayers having different surface charge as a function of outermost layer. **c**, Specific data sheet of $(\text{GOx}/\text{TREN})_m$ multilayers.

In summary, the electron transport mechanism of the $(\text{GOx}/\text{TREN})_m$ multilayer-coated MCFs without electron mediator basically follows the electron tunneling mechanism between GOx and metal electrode. Although the bilayer number (m) of $(\text{GOx}/\text{TREN})_m$ multilayers was increased up to

20 and 30, the thickness of $(\text{GOx}/\text{TREN})_m$ multilayers deposited onto the outer surface of MCFs was measured to be $\sim 9 \pm 3$ nm (for 20 bilayers) and $\sim 13 \pm 5$ nm (for 30 bilayers). **Considering the dimension of GOx ($6.0 \times 5.2 \times 7.7$ nm³), these thickness range of the multilayers indicate that GOx enzymes are adsorbed on the substrates as single or double layers.** However, in the case of the GOx/TREN multilayers onto the interior of highly porous MCFs, it is considered that their thicknesses will be much thinner with going from the outer surface to the center region of MCFs because the repetitive deposition of bulky GOx layers can block-up the porous structure of MCFs. Therefore, the overall anodic current density of MCF-based anode can be gradually increased up to the thickness of GOx/TREN multilayers allowing efficient electron tunneling. Additionally, in view of the structural frame, highly porous MCFs with the large surface area can significantly decrease electron transfer resistance compared to other nonporous electrodes, which can consequently boost up the power performance of metallic cotton-based biofuel cells.

However, to more clarify the issues raised by reviewer #1, we have added new Supplementary Figures 11, 14, 15 to the Supplementary Information and the text was newly included on page 9~11 in the revised manuscript as follows.

On the revised Supplementary Information:

Supplementary Figure 11. Cross-sectional FE-SEM images for film thickness of **a**, $(\text{GOx}/\text{TREN})_{20}$ and **b**, $(\text{GOx}/\text{TREN})_{30}$ multilayers deposited onto the Au-coated Si wafer.

Supplementary Figure 14. Nyquist plots of the $(\text{GOx}/\text{TREN})_{20}$ multilayers coated onto MCF (red solid circle) and nonporous plate substrate (i.e., Au-coated Si wafer substrate, blue solid square). In this case, ESR values of each electrode were measured to be $\sim 38.6 \Omega$ for 20-GOx/MCF and $\sim 62.6 \Omega$ for 20-GOx/Au-coated Si wafer electrodes, respectively.

^{a)}The interfacial electron transfer resistance between the charged interface of the electrode and the electrolyte.

Supplementary Figure 15. Outermost layer-dependent interfacial electron transfer kinetics (tested on the Au-coated Si wafer substrate). **a**, Nyquist plots of $(\text{GOx}/\text{TREN})_m$ multilayers as a function of bilayer number (m). **b**, Representative schematics of $(\text{GOx}/\text{TREN})_m$ multilayers having different surface charge as a function of outermost layer. **c**, Specific data sheet of $(\text{GOx}/\text{TREN})_m$ multilayers.

On page 9 in the revised manuscript: “Additionally, the formed $(\text{GOx}/\text{TREN})_m$ multilayers, which are sequentially deposited with small-molecule linkers instead of conventional bulky polyelectrolytes^{35–37}, exhibited the ultrathin film thicknesses of $\sim 9 \pm 3$ nm for 20 bilayers and $\sim 13 \pm 5$

nm for 30 bilayers (**Supplementary Fig. 11**). Considering the dimension of GOx ($6.0 \times 5.2 \times 7.7 \text{ nm}^3$)⁴⁸, these multilayer thicknesses indicate that (GOx/TREN)₂₀ and (GOx/TREN)₃₀ are adsorbed on the substrates like a single GOx layer and double GOx layers, respectively. As a result, the adsorption conformation of thin and uniform (GOx/TREN)_m multilayers can facilitate the electron transfer through the enzyme layers.”

On page 9 in the revised manuscript: “This bilayer (*m*)-dependent anodic performance can be explained by different adsorption behavior of the GOx layer on the inner and outer surface of the MCFs. In other words, in the case of the GOx/TREN multilayers onto the interior of highly porous MCFs, it is considered that their thicknesses will be much thinner with going from the outer surface to the center region of MCFs because the repetitive deposition of bulky GOx layers can block up the porous structure of MCFs. Therefore, the overall anodic current density of MCF-based anode can be gradually increased up to the thickness of GOx/TREN multilayers allowing efficient electron tunneling.”

On page 10 in the revised manuscript: “However, the ESR value of the MCF-based anode electrode is much lower than that of the nonporous substrate (i.e., Au-coated Si wafer)-based anode electrode (**Supplementary Fig. 14**) although the loading amount of (GOx/TREN)₃₀ multilayers onto MCF electrode is approximately 185 times higher than that of (GOx/TREN)₃₀ multilayers onto nonporous electrode (i.e., $\sim 869.5 \mu\text{g cm}^{-2}$ for MCF-based anode and $\sim 4.7 \mu\text{g cm}^{-2}$ for nonporous substrate-based anode at 30 bilayers), which indicate the effective electrode conformation of the MCF-based anode for facile electrochemical kinetics.”

On page 10 in the revised manuscript: “We also observed that the electron transfer kinetics at the interface of the electrode are significantly influenced by a kind of outermost layer of GOx/TREN multilayers (**Supplementary Fig. 15**). More specifically, OH-functionalized glucose has a high affinity (mainly, hydrogen-bonding interaction) with amine-functionalized TREN than negatively charged GOx at pH 7.4 [due to the presence of carboxylate ion (COO⁻) groups of GOx]. In line with this high affinity, glucose could be easily diffused into the outermost TREN-coated multilayer films, and therefore the outermost TREN-coated electrode exhibited much lower electron transfer resistance than the outermost GOx-coated electrodes in spite of the increase of layer number. These similar phenomena were also demonstrated by Willner group⁴⁹. Considering that all the biofuel cells shown in our study are based on the outermost TREN-coated electrodes as well as the highly porous electrodes, the electron transfer resistances of MCF-based electrodes can be significantly lowered compared to those of other nonporous electrodes.”

On references:

48. Libertino, S., Aiello, V., Scandurra, A., Renis, M. & Sinatra, F. Immobilization of the Enzyme Glucose Oxidase on Both Bulk and Porous SiO₂ Surfaces. *Sensors* **8**, 5637-5648 (2008).
49. Pardo-Yissar, V., Katz, E., Lioubashevski, O. & Willner, I. Layered polyelectrolyte films on Au electrodes: characterization of electron-transfer features at the charged polymer interface and application for selective redox reactions. *Langmuir* **17**, 1110–1118 (2001).

Comment 4. *The output power obtained in this work, 3.7 mWcm⁻², is not very special. There are papers reporting the power in the range of mW/cm². ECS Trans. 2009 16(38): 9-15*

Response 4: We appreciate the reviewer #1's comment. Although the previous paper by Hatazawa group [*ECS Trans.* **16**(38), 9-15 (2009)] reported the power density in the range of mW cm⁻², in sodium phosphate buffer, 1 M, pH 7, they used mediated electron-transfer BFC system with membrane film using electron mediator (2-amino-1,4-naphthoquinone). In general, the use of the electron (redox) mediator in BFC system can provide high power output by improving the electron transfer kinetics, it also causes several negative issues associated with toxicity and stability.

Additionally, unlike our micro-sized implantable fiber-type MCF-BFCs, they used carbon fiber sheet (1 cm x 1 cm, 750- μ m thickness) for anode and cathode electrode. These large-sized BFC electrodes have difficulties in terms of the practical implantable devices.

On the other hand, our BFC is membrane-less system and does not use any redox mediators for BFC system as already mentioned in the introduction part of our original manuscript. In general, a membrane-less BFCs have advantages such as small size for implantation into living organisms, long-term stability, and low cost, compared to the membrane-based BFCs. Additionally, to our knowledge, the resulting output power density of 3.7 mW cm⁻² in our membrane-less and mediator-less BFC is the best record obtained from the direct power measurement of current flow through an external resistor, which is approximately 36 times higher than the previously-reported best stationary power density (\sim 0.1 mW cm⁻²) of membrane-less and mediator-less BFC (tested under oxygen saturated condition, pH 3) by Prof. Yongchai Kwon's group [*Sci. Rep.*, **6**:30128 (2016)].

If the power density of MCF-based biofuel cells shown in our study was measured by using polarization tests, the output power density could be dramatically increased up to 8.7 mW cm⁻² due to the generation of capacitive and other parasite currents (see **Figure R6**).

Figure R6. Power output of the *n*-MCF-dependent MCF-BFC as a function of potential in 300 mmol l⁻¹ glucose under ambient conditions obtained by polarization tests.

Furthermore, as we described earlier, we would like to highlight that these output performances can be further enhanced by simply increasing the layer number (or mass) of active materials on the highly porous cellulose fibers. This possibility was confirmed by the change of the BFC performance with

the variation of bilayer number (n) of (TOA-Au NP/TREN) $_n$ multilayers coated on the cellulose fiber (*i.e.*, cathode electrode) (**Figures 5a, b, and d** in the revised manuscript). Thus, we believe that our results can provide a new platform for designing the BFC electrode conformation with high performance.

Figure 5. Enhancement of the power output of a MCF-BFC. **a**, Complete MCF-BFCs with a constant anode (30-GOx/20-MCF) and cathodes with a controlled number of bilayers n (n -MCF). **b**, Cathodic current curves for n -MCF cathodes as a function of n . **d**, Power output of the MCF-BFCs (an n -MCF cathode and a 30-GOx/20-MCF anode) with external variable resistors ($1 \text{ k}\Omega \sim 10 \text{ M}\Omega$) as a function of n -MCF.

For more clarifying this issue raised by Reviewer #1, we have now included the suggested two references regarding the BFC with the power density in the range of mW cm^{-2} [*ECS Trans.* 16, 9–15 (2009)] (Supplementary Reference 18) and the previously-reported best stationary power density of membrane-less and mediator-less BFC (tested under oxygen saturated condition, pH 3) by Prof. Yongchai Kwon’s group [*Sci. Rep.* 6, 30128 (2016)] (Supplementary Reference 19) in the revised Supplementary Information, and the new text is added in the revised manuscript as follows.

On the revised Supplementary Information:

Supplementary Table 1. Power performance and stability of various BFCs. Power density and operational stability of various BFCs reported to date.

Host electrode	Catalysts ^{b)} Anode/Cathode	Fuel Anode/Cathode	Operation condition	P _{max} (mW cm ⁻²)	OCV (V)	Ref.
MCF ^{a)}	GOx/Au NP	Glucose/O ₂	PBS (pH 7.4) at 37° C, ambient	3.7	-0.9	Our work
CNT fiber	GOx/BOD	Glucose/O ₂	PBS (pH 7.0) at 37 °C, air	0.74	0.83	4
CNT yam	GOx/BOD	Glucose/O ₂	PBS (pH 7.2) at 37 °C, O ₂	2.18	0.7	5
CNT film	GOx/Pt _{bulk} ^{c)}	Glucose/O ₂	PBS (pH 7.4), air	1.34	-0.8	6
Compressed CNTs	GOx-Cat/Lac	Glucose/O ₂	PBS (pH 7.0)	1.25	0.95	7
3D Au NP (on carbon paper)	FDH/BOD	Fructose/O ₂	0.1 M acetate buffer (pH 6.0) at 25 °C, O ₂	0.87	-0.7	8
SWNT (on glassy carbon)	GDH/Lac	Glucose/O ₂	PBS (pH 6) at 20 °C, ambient	0.0095	0.8	9
Graphene/Au NP Hybrid (on Au substrate)	FDH/Lac	Formic acid/O ₂	PBS (pH 6.0)	1.96	0.95	10
Nafion/poly(vinyl pyrrolidone) compound nanowire (on Au electrode)	GOx/Lac	Glucose/O ₂	PBS (pH 7.0)	0.03	0.23	11
Catecholamine polymers (on Au disk)	GOx/Carbon rod	Glucose/O ₂	PBS (pH 7.0)/ KMnO ₄ + H ₂ SO ₄	1.62	1.09	12
Au NP/PANI (polyaniline) network (on glassy carbon)	GOx/Lac	Glucose/O ₂	PBS (pH 7.4) at 20 °C	0.685	0.76	13
Enzyme cluster composite (on carbon paper)	GOx/Pt _{bulk} ^{c)}	Glucose/O ₂	1.0 M PBS (pH 7.4)	1.62	-0.8	14
Compressed MWNT ^{d)} /enzyme composite	GOx/Lac	Glucose/O ₂	In the abdominal cavity of a rat	0.193	0.57	15
Graphene and SWNT ^{d)} cogel	GOx/BOD	Glucose/O ₂	Air-saturated sodium phosphate buffer (0.1 M, pH 7.0) with 100 mM glucose	0.19	0.61	16
CS/CNC film (on glassy carbon)	GOx/Pt sheet	Glucose/O ₂	Air-saturated PBS (pH 7.2) containing 10 mM glucose	0.06	0.59	17
Carbon fiber sheets with membrane film using ANQ ^{e)}	GDH/BOD	Glucose/O ₂	Sodium phosphate buffer (1 M, pH 7)	3.0	0.5	18
(GOx or Lac)/PEI/CNT composite consisting of GA ^{f)}	GOx/Lac	Glucose/O ₂	PBS (pH 3), O ₂	0.102	-0.43	19

^{a)} MCF: metallic cotton fiber

^{b)} The abbreviations for different enzymes: glucose oxidase (GOx), laccase (Lac), bilirubin oxidase (BOD), fructose dehydrogenase (FDH), glucose dehydrogenase (GDH), and catalase (Cat).

^{c)} Pt_{bulk}: Commercial Pt bulk electrode

^{d)} MWNT: multi-walled carbon nanotube, SWNT: single-walled carbon nanotube, CS/CNC: chitosan/carbon nanochips

^{e)} 2-amino-1,4-naphthoquinone (ANQ) as an electron mediator

^{f)} PEI: poly(ethyleneimine), GA: glutaraldehyde

On page 15 in the revised manuscript: “As a result, our MCF-BFC composed of a 120-MCF cathode and a 30-GOx/20-MCF anode (i.e., 120-MCF-BFC) exhibited the maximum areal power density of 3.7 mW cm⁻² in 300 mmol l⁻¹ glucose under ambient condition through an external variable resistor, considerably outperforming conventional BFCs (including MET-BFCs) reported to date (Supplementary Table 1).”

On references,

Supplementary Reference 18. Sakai, H. et al. A high-power glucose/oxygen biofuel cell operating under quiescent condition. *ECS Trans.* **16**, 9–15 (2009).

Supplementary Reference 19. Christwardana, M., Kim, K. J. & Kwon, Y. Fabrication of Mediatorless/Membraneless glucose/oxygen based biofuel cell using biocatalysts including glucose oxidase and laccase enzymes. *Sci. Rep.*, **6**, 30128 (2016).

Reviewer #2

Overall comments to the Author: *The authors report here the development of a hybrid biofuel cells that employ a glucose oxidase anode, made of metallic cotton fibres, and a cathode made of the same electrode.*

Overall the manuscript is interesting and very well presented. Particularly appreciated are the characterisation made on the electrodes, both material characterisation and electrochemical characterisation, which provides a very comprehensive and systematic study. As such, this Reviewer suggests publications, after major revision.

One aspect the authors should focus on and enhance is the part related to the fuel cell operation, as most of the manuscript focuses on the development and characterisation of the electrodes only.

the results should be better discussed with reference on existing literature and particularly mediator-free enzymatic fuel cells.

It is noted that the graphs related to the electrochemical characterisation of the electrodes and of the fuel cells, have no error bars. Have the authors done any replicates of the experiments? what is the reproducibility of the results?

Can the enzymatic loading be estimated? considering that the immobilisation relies on an electrostatic interaction, how reproducible electrodes are and how stable? Have the authors experienced any leaching over prolonged operation?

Was the fuel cell performance assessed only via polarisation tests or was the voltage versus time recorded for long period of times?

Does Figure 22c refer to peak power values obtained from polarisation tests or does it refer to continuous operation in buffer? For practical application tests on performance under continuous operation is key.

Comment 1. *One aspect the authors should focus on and enhance is the part related to the fuel cell operation, as most of the manuscript focuses on the development and characterisation of the electrodes only. the results should be better discussed with reference on existing literature and particularly mediator-free enzymatic fuel cells.*

Response 1: We appreciate the reviewer #2's valuable comment on improving the manuscript. According to the reviewer #2's comments, we have modified the main text and included the suggested recent three references regarding the mediator-free enzymatic fuel cells (see Supplementary References 15-17). As checked in the Supplementary References 15-17, mediator-free enzymatic fuel cells generally provided relatively low power output generally, $\sim 1 \text{ mW cm}^{-2}$ or smaller than that value, measured by an external variable resistor), compared to the mediated electron transfer BFCs. We have additionally included the suggested reference regarding the best previously-reported stationary power density of membrane-less and mediator-less BFC (tested under oxygen saturated condition, pH 3) by Prof. Yongchai Kwon's group [*Sci. Rep.*, **6**, 30128 (2016)] (Supplementary Reference 19 in the Supplementary Table 1).

For more clarifying this issue raised by Reviewer #2, we have additionally added the new discussion especially for the mediator-free enzymatic fuel cells in the modified Supplementary Table 1, and new text related to the fuel cell operation were included in the revised manuscript and Supplementary Information as follows.

On the revised Supplementary Information (with regard to the suggested references for mediator-free enzymatic fuel cells, Supplementary References 15, 16, 17):

Supplementary Reference 15. Zebda, A. et al. Single glucose biofuel cells implanted in rats power electronic devices. *Sci. Rep.* **3**, 1516 (2013).

Supplementary Reference 16. Campbell, A. S. et al. Membrane/Mediator-free rechargeable enzymatic biofuel cell utilizing graphene/single-wall carbon nanotube cogel electrodes. *ACS Appl. Mater. Interfaces* **7**, 4056–4065 (2015).

Supplementary Reference 17. Kang, Z. et al. Direct electrochemistry and bioelectrocatalysis of glucose oxidase in CS/CNC film and its application in glucose biosensing and biofuel cells. *RSC Adv.* **7**, 4572–4579 (2017).

Supplementary Table 1. Power performance and stability of various BFCs. Power density and operational stability of various BFCs reported to date.

Host electrode	Catalysts ^{b)} Anode/Cathode	Fuel Anode/Cathode	Operation condition	P _{max} (mW cm ⁻²)	OCV (V)	Ref.
MCF ^{a)}	GOx/Au NP	Glucose/O ₂	PBS (pH 7.4) at 37 °C, ambient	3.7	-0.9	Our work
CNT fiber	GOx/BOD	Glucose/O ₂	PBS (pH 7.0) at 37 °C, air	0.74	0.83	4
CNT yarn	GOx/BOD	Glucose/O ₂	PBS (pH 7.2) at 37 °C, O ₂	2.18	0.7	5
CNT film	GOx/Pt _{bulk} ^{c)}	Glucose/O ₂	PBS (pH 7.4), air	1.34	-0.8	6
Compressed CNTs	GOx-Cat/Lac	Glucose/O ₂	PBS (pH 7.0)	1.25	0.95	7
3D Au NP (on carbon paper)	FDH/BOD	Fructose/O ₂	0.1 M acetate buffer (pH 6.0) at 25 °C, O ₂	0.87	-0.7	8
SWNT (on glassy carbon)	GDH/Lac	Glucose/O ₂	PBS (pH 6) at 20 °C, ambient	0.0095	0.8	9
Graphene/Au NP Hybrid (on Au substrate)	FDH/Lac	Formic acid/O ₂	PBS (pH 6.0)	1.96	0.95	10
Nafion/poly(vinyl pyrrolidone) compound nanowire (on Au electrode)	GOx/Lac	Glucose/O ₂	PBS (pH 7.0)	0.03	0.23	11
Catecholamine polymers (on Au disk)	GOx/Carbon rod	Glucose/O ₂	PBS (pH 7.0)/ KMnO ₄ + H ₂ SO ₄	1.62	1.09	12
Au NP/PANI (polyaniline) network (on glassy carbon)	GOx/Lac	Glucose/O ₂	PBS (pH 7.4) at 20 °C	0.685	0.76	13
Enzyme cluster composite (on carbon paper)	GOx/Pt _{bulk} ^{c)}	Glucose/O ₂	1.0 M PBS (pH 7.4)	1.62	-0.8	14
Compressed MWNT ^{d)} /enzyme composite	GOx/Lac	Glucose/O ₂	In the abdominal cavity of a rat	0.193	0.57	15
Graphene and SWNT ^{d)} cogel	GOx/BOD	Glucose/O ₂	Air-saturated sodium phosphate buffer (0.1 M, pH 7.0) with 100 mM glucose	0.19	0.61	16
CS/CNC film (on glassy carbon)	GOx/Pt sheet	Glucose/O ₂	Air-saturated PBS (pH 7.2) containing 10 mM glucose	0.06	0.59	17
Carbon fiber sheets with membrane film using ANQ ^{e)}	GDH/BOD	Glucose/O ₂	Sodium phosphate buffer (1 M, pH 7)	3.0	0.5	18
(GOx or Lac)/PEI/CNT composite consisting of GA ^{f)}	GOx/Lac	Glucose/O ₂	PBS (pH 3), O ₂	0.102	-0.43	19

a) MCF: metallic cotton fiber

b) The abbreviations for different enzymes: glucose oxidase (GOx), laccase (Lac), bilirubin oxidase (BOD), fructose dehydrogenase (FDH), glucose dehydrogenase (GDH), and catalase (Cat).

c) Pt_{bulk}: Commercial Pt bulk electrode

d) MWNT: multi-walled carbon nanotube, SWNT: single-walled carbon nanotube, CS/CNC: chitosan/carbon nanochips

e) 2-amino-1,4-naphthoquinone (ANQ) as an electron mediator

f) PEI: poly(ethyleneimine), GA: glutaraldehyde

On page 15 in the revised manuscript: “This high-power output (3.7 mW cm⁻²) obtained from our membrane-free and mediator-free MCF-BFC is in stark contrast with those measured from other mediator-free enzymatic fuel cells (i.e., in the case of being measured by an external variable resistor, the power output of mediator-free enzymatic fuel cells is generally less than ~1 mW cm⁻²). For examples, the power output shown in our system is nearly three times higher than that of the previous DET-based compressed CNT electrode⁵ (1.25 mW cm⁻², measured by polarization tests (i.e., CV

measurements) without an external variable resistor), and even higher than that of the reported osmium complex-based bisrolled carbon nanotube yarn MET-BFC⁷ (2.18 mW cm⁻²). Consequently, we demonstrated that our highly porous metallic cotton fiber-based BFC (MCF-BFC) fabricated by small molecule linker-induced LbL assembly of the active catalysts can significantly realize the DET rate between the enzymes and the conductive supports.”

On page 9 in the revised manuscript: “This bilayer (*m*)-dependent anodic performance can be explained by different deposition behavior of the GOx layer on the inner and outer surface of the MCFs. In other words, in the case of the GOx/TREN multilayers onto the interior of highly porous MCFs, it is considered that their thicknesses will be much thinner with going from the outer surface to the center region of MCFs because the repetitive deposition of bulky GOx layers can block up the porous structure of MCFs. Therefore, the overall anodic current density of MCF-based anode electrode can be gradually increased up to the thickness of GOx/TREN multilayers allowing efficient electron tunneling.”

On page 13 in the revised manuscript (for the enhancement related to the anode performance): “Additionally, we expect that the anodic performance can be further controlled and enhanced with increasing the electrical conductivity of MCF for electrical communication and the surface area of MCF for efficient enzyme loading (by a further increase (*n* > 20) of bilayer number of (TOA-Au NP/TREN)_{*n*} multilayers onto cotton fibers).”

On page 9-10 in the revised manuscript (regarding operational stability of the enzymatic anode electrode in Supplementary Figure 13): “In this case, the 30-GOx/20-MCF anode maintained ~97.5 % of its initial current density after continuous operation for 1 hour (herein, the initial current density of the anode electrode was obtained after pre-training for 1 day in 300 mmol l⁻¹ glucose containing PBS buffer condition), indicating the excellent electrode stability caused by the precisely controlled deposition of GOx using TREN (Supplementary Fig. 13).”

On the revised Supplementary Figure 13:

Supplementary Figure 13. Operational stability of anodic current density (%) for (GOx/TREN)₃₀ multilayers.

On page 10 in the revised manuscript (for internal resistance shown in Supplementary Figure 14):

“However, the ESR value of the MCF-based anode electrode is much lower than that of the nonporous substrate (i.e., Au-coated Si wafer)-based anode electrode (**Supplementary Fig. 14**) although the loading amount of (GOx/TREN)₃₀ multilayers onto MCF electrode is approximately 185 times higher than that of (GOx/TREN)₃₀ multilayers onto nonporous electrode (i.e., ~860 $\mu\text{g cm}^{-2}$ for MCF-based anode and ~4.65 $\mu\text{g cm}^{-2}$ for nonporous substrate-based anode at 30 bilayers), which indicate the effective electrode conformation of the MCF-based anode for facile electrochemical kinetics.”

On the revised Supplementary Information:

Supplementary Figure 14. Nyquist plots of the (GOx/TREN)₂₀ multilayers coated onto MCF (red solid circle) and nonporous plate substrate (i.e., Au-coated Si wafer substrate, blue solid square). In this case, ESR values of each electrode were measured to be ~38.6 Ω for 20-GOx/MCF and ~62.6 Ω for 20-GOx/Au-coated Si wafer electrodes, respectively.

On Page 10 in in the revised manuscript (for interfacial resistance shown in Supplementary Figure 15):

“We also observed that the electron transfer kinetics at the interface of the electrode are significantly influenced by a kind of outermost layer of GOx/TREN multilayers (**Supplementary Fig. 15**). More specifically, OH-functionalized glucose has a high affinity (mainly, hydrogen-bonding interaction) with amine-functionalized TREN than negatively charged GOx at pH 7.4 [due to the presence of carboxylate ion (COO^-) groups of GOx]. In line with this high affinity, glucose could be easily diffused into the outermost TREN-coated multilayer films, and therefore the outermost TREN-coated electrode exhibited much lower electron transfer resistance than the outermost GOx-coated electrodes in spite of the increase of layer number. These similar phenomena were also demonstrated by Willner group⁴⁹. Considering that all the biofuel cells shown in our study are based on the outermost TREN-coated electrodes as well as the highly porous electrodes, the electron transfer

resistances of MCF-based electrodes can be significantly lowered compared to those of other nonporous electrodes.”

On the revised supplementary Figure 15:

c

n	Terminated with	ESR (Ω)	$R_{et}^{a)}$ (Ω)
20	TREN	62.5	63
20.5	GOx	62.8	757
21	TREN	63.8	64
21.5	GOx	67.1	1415
22	TREN	66.1	66
22.5	GOx	66.8	1836
23	TREN	74.9	75

^{a)}The interfacial electron transfer resistance between the charged interface of the electrode and the electrolyte.

Supplementary Figure 15. Outermost layer-dependent interfacial electron transfer kinetics (tested on the Au-coated Si wafer substrate). **a**, Nyquist plots of $(GOx/TREN)_m$ multilayers as a function of bilayer number (m). **b**, Representative schematics of $(GOx/TREN)_m$ multilayers having the different surface charge as a function of outermost layer. **c**, Specific data sheet of $(GOx/TREN)_m$ multilayers.

On page 12-13 in the revised manuscript (for the catalytic kinetics shown in Supplementary Figure 21): “Here, small ΔE_p (60 mV at a scan rate of 100 mV s⁻¹) indicates fast heterogeneous electron transfer process, and the reaction is not a diffusion-controlled process but a surface-controlled one, as expected for immobilized system (more specifically, in the case of the surface-controlled system, ΔE_p is less than 200 mV⁵⁴). Additionally, a large k_s of 6.0 ± 0.1 s⁻¹ obtained from 30-GOx/20-MCF suggests that the electrode is significantly effective in facilitating the direct electron transfer of GOx⁵⁵⁻⁵⁷. As a result, the formed 30-GOx/20-MCF electrode provides efficient catalytic kinetics with smaller ΔE_p and larger k_s . (for examples, 1.53 s⁻¹ for multi-walled carbon nanotubes (MWNT)-modified electrode, 0.3 s⁻¹ for single-walled carbon nanotubes (SWNT)-modified electrode, 3.96 s⁻¹ for the porous TiO₂ electrode, and 6.28 s⁻¹ for GOx-carbon nanodots on the glassy carbon electrode)⁵⁸”

On the revised supplementary Figure 21:

Supplementary Figure 21. Electrochemical response of the GOx-immobilized MCF anode electrode. **a**, Scan rate (ν)-dependent CVs of the anodic electrode, 30-GOx/20-MCF, in 300 mmol l^{-1} glucose buffer condition. Scan rates (from inner to outer of CV curves) were defined as 0.005, 0.01, 0.02, 0.05, 0.1, 0.2, 0.3, 0.4, 0.5, 0.7, and 1.0 V s^{-1} . **b**, The plot of peak currents of 30-GOx/20-MCF with increasing the scan rate from 0.005 to 1 V s^{-1} . **c**, The plot of potential ($E - E^0$) versus $\log(\nu)$.

On References:

54. Zhao, M., Gao, Y., Sun, J. & Gao, F. Mediatorless glucose biosensor and direct electron transfer type glucose/air biofuel cell enabled with carbon nanodots. *Anal. Chem.* **87**, 2615–2622 (2015).
55. Cai, C. & Chen, J. Direct electron transfer of glucose oxidase promoted by carbon nanotubes. *Anal. Biochem.* **332**, 75–83 (2004).
56. Hong, J. et al. Direct electrochemistry of hemoglobin immobilized on a functionalized multi-walled carbon nanotubes and gold nanoparticles nanocomplex-modified glassy carbon electrode. *Sensors* **13**, 8595–8611 (2013).
57. Hui, N., Gao, R., Li, X., Sun, W. & Jiao, K. Direct electrochemistry of hemoglobin immobilized in polyvinylalcohol and clay composite film modified carbon ionic liquid electrode. *J. Braz. Chem. Soc.* **20**, 252–258 (2009).
58. Bao, S. et al. New nanostructured TiO_2 for direct electrochemistry and glucose sensor applications. *Adv. Funct. Mat.* **18**, 591–599 (2008).

Supplementary Reference 15. Zebda, A. et al. Single glucose biofuel cells implanted in rats power electronic devices. *Sci. Rep.* **3**, 1516 (2013).

Supplementary Reference 16. Campbell, A. S. et al. Membrane/Mediator-free rechargeable

enzymatic biofuel cell utilizing graphene/single-wall carbon nanotube cogel electrodes. *ACS Appl. Mater. Interfaces* 7, 4056–4065 (2015).

Supplementary Reference 17. Kang, Z. et al. Direct electrochemistry and bioelectrocatalysis of glucose oxidase in CS/CNC film and its application in glucose biosensing and biofuel cells. *RSC Adv.* 7, 4572–4579 (2017).

Comment 2. It is noted that the graphs related to the electrochemical characterisation of the electrodes and of the fuel cells, have no error bars. Have the authors done any replicates of the experiments? what is the reproducibility of the results?

Response 2: We appreciate the reviewer #2’s comment. In the present work, all of our reported results were obtained from reproducible average values of three or five replicated samples in each experiment conditions.

To clarify this issue raised by reviewer #2, we have added the error bars with the calculation of the relative standard deviation. This information has been now added to the captions of each figure as follows:

On the caption of Figure 2e in the revised manuscript: “The error bars show the standard deviation from the mean value of electrical conductivities for three to five independent experiments.”

On the caption of Figure 3c in the revised manuscript: “The error bars show the standard deviation from the mean value of current densities for three to five independent experiments.”

On the caption of Figure 5d,5e in the revised manuscript: “The error bars (d, e) show the standard deviation from the mean value of power densities for three to five independent experiments.”

On Figure 3c, 5d, and 5e of the revised manuscript:

Inset of Figure 3c

Figure 5d

Figure 5e

On the caption of Supplementary Figure 26b in the revised manuscript: “The error bars show the standard deviation from the mean value of power densities for three to five independent experiments.”

On the caption of Supplementary Figure 29a, 29b in the revised manuscript: “All error bars show the standard deviation from the mean value of power densities for three to five independent experiments.”

On the revised Supplementary Information:

Supplementary Figure 26b

Supplementary Figure 29a

Supplementary Figure 29b

Comment 3. *Can the enzymatic loading be estimated? considering that the immobilisation relies on an electrostatic interaction, how reproducible electrodes are and how stable? Have the authors experienced any leaching over prolonged operation?*

Response 3: We appreciate the reviewer #2’s comment. We address each in turn below.

3-1. With regard to loading amount of the enzyme (GOx).

Actually, it is not easy to measure the exact loading amount of GOx and TREN separately because the LbL-assembled GOx/TREN multilayers were distributed evenly throughout the interior and outer surface of the cellulose fiber [see **Figure 4a** (right side) and **Supplementary Figure 22c,d** in the revised manuscript and Supplementary Information]. However, given that TREN molecule has a very low M_w (~146 g mol⁻¹) compared to bulky GOx with M_w of ~160 kDa, it is assumed that the loading amount of GOx in the multilayers is dominant.

In the case of 30-bilayered GOx/TREN multilayers, the thickness was measured to be $\sim 13 \pm 5$ nm (please see **Figure R7**). We also quantitatively measured the loading amount of the GOx/TREN multilayers coated onto the MCF using an analytical balance (XP205 model, Mettler Toledo, resolution of 0.01 mg). As a result, the loading amount of (GOx/TREN)₃₀ multilayers was measured to

be approximately $860 \mu\text{g cm}^{-2}$ ($\sim 408 \pm 35 \mu\text{g}$), which was ~ 185 times higher than that of the nonporous substrate (*i.e.*, Au-coated Si wafer)-based one ($\sim 4.65 \mu\text{g cm}^{-2}$) at the same 30 bilayers.

On the revised manuscript:

Figure 4. a, SEM image and illustrated top view of $(\text{GOx/TREN})_{30}$ multilayer-coated MCFs.

On the revised Supplementary Information:

Supplementary Figure 22. c, d, FE-SEM images of the (30-GOx/20-MCF) anodes incorporating TREN linkers [$(\text{GOx/TREN})_{30}$ on-MCF].

Figure R7. Cross-sectional FE-SEM images for film thickness of (GOx/TREN)₃₀ multilayers deposited onto the Au-coated Si wafer.

3-2. With regard to reproducibility and stability of the electrodes.

Our BFC electrodes were prepared by electrostatic LbL assembly of the negatively charged GOx (5 mg ml⁻¹ in PBS for 10 min) and the positively charged TREN (1 mg ml⁻¹ in PBS for 10 min), at pH 7.4 PBS solution. All the experiments were carried out at least three or five times using independent sample electrodes under same experimental conditions, and the results were selected the reasonable average values (*i.e.*, standard deviation) for each analysis (*more details can be found in Experimental Section in the main manuscript*). In order to clarify the reproducibility and stability of our enzymatic anode electrodes, we have tested time-dependent anodic current density change [I/I_0 (%)] for 1 hour (*Here, the initial current density of the anode electrode was obtained after pre-training for 1 day in 300 mmol l⁻¹ glucose containing PBS buffer condition*) for (GOx/TREN)₃₀/20-MCF (see **Figure R9**) (*here, the results obtained indicate the average values from three different electrodes in each case*). As shown in **Figure R8**, the (GOx/TREN)₃₀/20-MCF showed the excellent operation stability of ~97.5% of initial current density with continuous operation. This high operation stability of TREN-based BFCs can be explained by the role of amine-functionalized molecule linker (*i.e.*, TREN). In this case, TREN plays important roles in not only reducing the inter-distance and contact resistance between the introduced materials (*i.e.*, GOx/GOx, GOx/Au NP, and Au NP/Au NP) but also providing the stable bonding of each materials by electrostatic and/or covalent bonds, resulting in the desirable operation stability and output performances. This is quite different from the conventional BFC electrode conformation with bulky and thick single enzyme layer, resulting in the low activation efficiency despite the high mass loading of GOx due to their insulating nature and self-aggregation.

Figure R8. Stability of anodic current density (%) for (GOx/TREN)₃₀/20-MCF anode electrodes.

3-3. With regard to the leaching issue over long-term operation.

According to the long-term stability test shown in the above-mentioned **Figure R9**, we confirmed that there was a leaching problem of the enzyme in the enzymatic anode electrode. However, the anodic current density measured for 1 hr was stably operated with a slight decrease from the initial status

with a sufficient pre-training step (~1 day). For more confirmation, the complete BFC cell prepared by connecting the anode (*i.e.*, 30-GOx/20-MCF) and cathode (*i.e.*, 120-MCF) electrode was continuously operated in 10 mmol l⁻¹ glucose buffer solution for prolonged operation time. These results are shown in **Figure R9**.

Figure R9. a. Relative voltage retention (OCV/OCV_0) of the complete MCF-BFC, which is measured under open-circuit system. **b.** Relative power retention (P/P_0) of the complete MCF-BFC in 10 mmol l⁻¹ glucose buffer during 35 days. In this case, the power output changes of MCF-BFCs with external variable resistors were continuously measured as a function of time.

However, to more clarify the issues raised by reviewer #2, we have additionally added the discussion and Figure R8 and R9 to our revised manuscript and Supplementary Information as follows.

On the revised Supplementary Information:

Supplementary Figure 11. b. $(GOx/TREN)_{30}$ multilayers deposited onto the Au-coated Si wafer.

On Supplementary Information:

Supplementary Figure 13. Stability of anodic current density (%) for (GOx/TREN)₃₀/20-MCF anode electrode.

On Supplementary Information:

Supplementary Figure 26. Performance of MCF-BFCs. **c**, Relative voltage retention (OCV/OCV_0) of the complete MCF-BFC, which is measured under the open-circuit system. **d**, Relative power retention (P/P_0) of the complete MCF-BFC in 10 mmol l⁻¹ glucose buffer for 35 days. In this case, the

power output changes of MCF-BFCs with external variable resistors were continuously measured as a function of time.

On Page 10 in the revised manuscript: “~the nonporous substrate (i.e., Au-coated Si wafer)-based anode electrode (**Supplementary Fig. 14**) although the loading amount of (GOx/TREN)₃₀ multilayers onto MCF electrode is 185 times higher than that of (GOx/TREN)₃₀ multilayers onto nonporous electrode (i.e., ~860 $\mu\text{g cm}^{-2}$ for MCF-based anode and ~4.65 $\mu\text{g cm}^{-2}$ for nonporous substrate-based anode at 30 bilayers), which indicate the effective electrode conformation of the MCF-based anode for facile electrochemical kinetics.”

On Page 9-10 in the revised manuscript: “In this case, the 30-GOx/20-MCF anode maintained ~97.5 % of its initial current density after continuous operation for 1 hour (*herein, the initial current density of the anode electrode was obtained after pre-training for 1 day in 300 mmol l⁻¹ glucose containing PBS buffer condition*), indicating the excellent electrode stability caused by the precisely controlled deposition of GOx using TREN (**Supplementary Fig. 13**).”

On Page 14 in the revised manuscript: “The MCF-BFC also exhibited high operational stability, maintaining ~71% (~0.6 mW cm⁻²) of its initial power density and voltage retention of ~83% after continuous operations for 20 days (~62% of its initial power density and voltage retention of ~79% after 35 days) (**Supplementary Fig. 26c and d**).”

Comment 4. Was the fuel cell performance assessed only via polarisation tests or was the voltage versus time recorded for long period of times?

Response 4: We appreciate the reviewer #2’s comment. In the present study, the power densities were obtained by operating the BFC through an external variable resistor to control the cell voltage and measure the current, instead of transient power density *via* polarization tests. These power densities were continuously measured for a long period of time, 35 days.

However, to more clarify these issues raised by reviewer 2, we have now revised the figure caption of our revised manuscript and additionally added the plot of open-circuit voltage (OCV) vs. time to the Supplementary Information of our revised manuscript as follows:

On the caption of Figure 5e: “In this case, the two MCF-BFCs were connected with each other in series, and its power output was also measured with an external variable resistor.”

On the caption of Supplementary Figure 26: “b, Power output of the complete BFC with external variable resistors (1 k Ω ~ 10 M Ω) consisting of the 30-GOx/20-MCF anode and 20-MCF cathode as a function of potential. c, Relative voltage retention (OCV/OCV_0) of the complete MCF-BFC, which is measured under the open-circuit system. d, Relative power retention (P/P_0) of the complete MCF-BFC in 10 mmol l⁻¹ glucose buffer for 35 days. In this case, the power output changes of MCF-BFCs with an external variable resistor were continuously measured as a function of time.”

On the caption of Supplementary Figure 29: “Characteristics of the MCF-BFCs by measuring the current flowing through external variable resistors (in the range of 1 kΩ ~ 10 MΩ) to control cell potential.”

On the revised Supplementary Information:

Supplementary Figure 26. Performance of MCF-BFCs. **c**, Relative voltage retention (OCV/OCV_0) of the complete MCF-BFC, which is measured under the open-circuit system. **d**, Relative power retention (P/P_0) of the complete MCF-BFC in 10 mmol l⁻¹ glucose buffer for 35 days. In this case, the power output changes of MCF-BFCs with external variable resistors were continuously measured as a function of time.

Comment 5. Does Figure 22c refer to peak power values obtained from polarisation tests or does it refer to continuous operation in buffer? For practical application tests on performance under continuous operation is key.

Response 5: We appreciate the reviewer #2’s comment. The peak power densities in **Supplementary Figure 22c** in the original version (please see **Figure R10** and **Supplementary Figure 26d** in the revised Supplementary Information). All power density values were determined by measuring the current flowing through an external variable resistor in the range of 1 kΩ ~ 10 MΩ to control cell voltage. Additionally, if the power density of MCF-based biofuel cells shown in our study was measured using polarization tests, the output power density could be increased up to 8.7 mW cm⁻² due to the generation of capacitive and other parasite currents, which was also tested with continuous operation in PBS buffer (see **Figure R11**). However, given that this ultrahigh power density value was not stationary power possibly used for practical applications but transient power, these power output results obtained from polarization tests were not included in our manuscript. We also strongly agree with the referees’ comments that biofuel cells should be continuously operated, and this is the essential condition for practical applications.

Figure R10. Relative power retention (P/P_0) of the complete MCF-BFC in 10 mmol l^{-1} glucose buffer during 35 days.

Figure R11. Power output of the n -MCF-dependent MCF-BFC as a function of potential in 300 mmol l^{-1} glucose under ambient conditions obtained by polarization tests.

To more clarify the issues raised by reviewer #2, we have additionally added the Figure R11 to our Supplementary Information as follows.

On the revised Supplementary Information:

Supplementary Figure 26. Performance of MCF-BFCs. **d**, Relative power retention (P/P_0) of the complete MCF-BFC in 10 mmol l^{-1} glucose buffer for 35 days. In this case, the power output changes of MCF-BFCs with external variable resistors were continuously measured as a function of time.

Reviewers' comments:

Reviewer #2 (Remarks to the Author):

I would like to start this review by congratulating with the Authors for the effort that they have put into the preparation of their rebuttal and the revised manuscript. Nevertheless, I share somehow the concerns of Reviewer #1 on the innovation that this manuscript brings and whether this is enough for a publication in Nature Communication.

In any cases, I have the following doubts that would like to be clarified:

- 1) How confident are the authors on the precision of measuring the film thickness of (GOx/TREN) by FE-SEM images analysis Wouldn't it be more appropriate to refer to an 'estimated' thickness?
- 2) I am still not clear on how the monitoring of the BFC performance was done. The authors state that the anode and cathode were connected to variable external resistors to keep the cell voltage constant. How was this control done? Was it with a potentiostat? A more common set-up in BFCs involves the use of a fixed external resistance and the monitoring of the cell potential/current output, see for example the work by Wang, by Zebda or by Di Lorenzo, to name but a few. Also was the cell operated in flow-through mode as in some works reported by the authors previously mentioned or in batch mode? In either case it would be interesting to see the evolution of the current versus time along one day and during the whole length of the experiment. finally, the supplementary figure 26 should give an indication on reproducibility of the data (error bars).

Response to Reviewers' Comments

Title: **High-Power Hybrid Biofuel Cells Using Layer-by-Layer Assembled Glucose Oxidase-Coated Metallic Cotton Fibers**

We appreciate the comments and suggestions raised by the reviewers. Our responses are listed below, and the main text and supplemental materials file have been modified according to the referees' comments.

REVIEWERS' COMMENTS:

Reviewer #2

Overall comments to the Author:

I would like to start this review by congratulating with the Authors for the effort that they have put into the preparation of their rebuttal and the revised manuscript. Nevertheless, I share somehow the concerns of Reviewer #1 on the innovation that this manuscript brings and whether this is enough for a publication in Nature Communication.

In any cases, I have the following doubts that would like to be clarified:

1) How confident are the authors on the precision of measuring the film thickness of (GOx/TREN) by FE-SEM images analysis Wouldn't it be more appropriate to refer to an 'estimated' thickness?

2) I am still not clear on how the monitoring of the BFC performance was done. The authors state that the anode and cathode were connected to variable external resistors to keep the cell voltage constant. How was this control done? Was it with a potentiostat? A more common set-up in BFCs involves the use of a fixed external resistance and the monitoring of the cell potential/current output, see for example the work by Wang, by Zebda or by Di Lorenzo, to name but a few. Also was the cell operated in flow-through mode as in some works reported by the authors previously mentioned or in batch mode? In either case it would be interesting to see the evolution of the current versus time along one day and during the whole length of the experiment. finally, the supplementary figure 26 should give an indication on reproducibility of the data (error bars).

Comment 1. *How confident are the authors on the precision of measuring the film thickness of (GOx/TREN) by FE-SEM images analysis Wouldn't it be more appropriate to refer to an 'estimated' thickness?*

Response 1: We appreciate reviewer #2's suggestion. Although we investigated the film thickness using cross-sectional FE-SEM images (**Supplementary Fig. 11**), it contained the experimental error on the precision of measuring each ultrathin film thickness for (GOx/TREN)₂₀ and (GOx/TREN)₃₀ multilayers. Therefore, the sentence related to the film thickness has been now changed in the revised manuscript as follows.

On page 9 in the revised manuscript: “Additionally, the film thickness of the formed (GOx/TREN)_m multilayers, which are sequentially deposited with small-molecule linkers instead of conventional bulky polyelectrolytes^{35–37}, was estimated to be $\sim 9 \pm 3$ and $\sim 13 \pm 5$ nm for 20 and 30 bilayers, respectively (**Supplementary Fig. 11**).”

Comment 2. *I am still not clear on how the monitoring of the BFC performance was done. The authors state that the anode and cathode were connected to variable external resistors to keep the cell voltage constant. How was this control done? Was it with a potentiostat? A more common set-up in BFCs involves the use of a fixed external resistance and the monitoring of the cell potential/current output, see for example the work by Wang, by Zebda or by Di Lorenzo, to name but a few. Also was the cell operated in flow-through mode as in some works reported by the authors previously mentioned or in batch mode? In either case it would be interesting to see the evolution of the current versus time along one day and during the whole length of the experiment. Finally, the supplementary figure 26 should give an indication on reproducibility of the data (error bars).*

2-1. With regard to the monitoring of the BFC performance.

Response 2-1: Thank you for reviewer's valuable comments regarding the monitoring of the BFC performance. The word “variable” used in the manuscript implied that a various range of external resistors were used to measure the performance of the BFCs. Therefore, the resistors used in our system are not the “changeable” resistors. That is, we measured the power densities of BFC with the use of a fixed external resistance. When the fixed external resistance was applied to the BFCs, the cell potential was independently controlled, and the current output and power densities were determined by measuring at a fixed resistance (in the range of 1 k Ω ~ 1 M Ω). For example, at 1 k Ω of external resistance, we measured each voltage and current, and then power density was determined according to the electric power formula ($P = V \times I$). Additionally, our complete BFC was tested in batch mode, not in flow-through mode. This BFC set-up is common for monitoring the current outputs and power densities by several research groups [Wang (**R1**), Zebda (**R2**) and Di Lorenzo (**R3**)].

- R1. Bandothkar, A. J. et al. Soft, stretchable, high power density electronic skin-based biofuel cells for scavenging energy from human sweat. *Energy Environ. Sci.* **10**, 1581–1589 (2017).
- R2. Zebda, A. et al. Mediatorless high-power glucose biofuel cells based on compressed carbon nanotube-enzyme electrodes. *Nat. Commun.* **2**, 370 (2011).
- R3. du Toit, H. & Di Lorenzo, M. Continuous power generation from glucose with two different miniature flow-through enzymatic biofuel cells. *Biosens. Bioelectron.* **69**, 199–205 (2015).

However, for more clarifying this issue raised by reviewer #2 and to avoid confusion, “external variable resistor” was replaced by “fixed external resistor” in the revised manuscript and modified the electrochemical measurements section in the Method, as follows:

On page 4 in the revised manuscript: “As a result, our hybrid MCF-BFC provides a remarkable power density of 3.7 mW cm^{-2} using a fixed external resistance without the aid of redox mediators, significantly exceeding the power density of the previously reported conventional DET- and MET-BFCs.”

On page 7 in the revised manuscript: “please note that the power density of MCF-based hybrid BFCs was evaluated by measuring the current output applying a fixed external resistance to eliminate the capacitive and other parasite currents.”

On page 14 in the revised manuscript: “To demonstrate a working MCF-BFC, the 30-GOx/20-MCF anode was connected to the 20-MCF cathode and the power density was evaluated by measuring the current flow through a fixed external resistance in the range of $1 \text{ k}\Omega$ to $10 \text{ M}\Omega$ without membrane (Supplementary Fig. 26a).”

On page 15 in the revised manuscript:

“The assembled cathodes with different numbers of bilayers were connected to a constant anode (i.e., 30-GOx/20-MCF) with an external resistor for the preparation of complete BFCs.”

“As a result, our MCF-BFC composed of a 120-MCF cathode and a 30-GOx/20-MCF anode (i.e., 120-MCF-BFC) exhibited the maximum areal power density of 3.7 mW cm^{-2} in 300 mmol l^{-1} glucose under ambient condition applying a fixed external resistance, considerably outperforming conventional BFCs (including MET-BFCs) reported to date (Supplementary Table 1).”

“This high-power output (3.7 mW cm^{-2}) obtained from our membrane-free and mediator-free MCF-BFC is in stark contrast with those measured from other mediator-free enzymatic fuel cells (i.e., in the case of being measured by an external resistor, the power output of mediator-

free enzymatic fuel cells is generally less than $\sim 1 \text{ mW cm}^{-2}$.”

“For examples, the power output shown in our system is nearly three times higher than that of the previous DET-based compressed CNT electrode⁵ (1.25 mW cm^{-2} , measured by polarization tests (i.e., CV measurements) **without an external resistor**), and even higher than that of the reported osmium complex-based biscrolled carbon nanotube yarn MET-BFC⁷ (2.18 mW cm^{-2}).”

On page 16 in the revised manuscript: “Consequently, the areal power density of 3.7 mW cm^{-2} obtained from the 120-MCF-BFC **using a fixed external resistance** exhibited the best-reported performance of the BFCs.”

On page 20 in the revised manuscript: “The BFC power densities were determined **by measuring the current output applying a fixed external resistance** (in the range of $1 \text{ k}\Omega \sim 10 \text{ M}\Omega$) to control cell potential^{3,5,66}.”

On references:

3. Bandodkar, A. J. et al. Soft, stretchable, high power density electronic skin-based biofuel cells for scavenging energy from human sweat. *Energy Environ. Sci.* **10**, 1581–1589 (2017).
5. Zebda, A. et al. Mediatorless high-power glucose biofuel cells based on compressed carbon nanotube-enzyme electrodes. *Nat. Commun.* **2**, 370 (2011).
66. du Toit, H. & Di Lorenzo, M. Continuous power generation from glucose with two different miniature flow-through enzymatic biofuel cells. *Biosens. Bioelectron.* **69**, 199–205 (2015).

On the caption of Figure 5d,e in the revised manuscript: “**d**, Power output of the MCF-BFCs (an *n*-MCF cathode and a 30-GOx/20-MCF anode) **with external resistors** ($1 \text{ k}\Omega \sim 10 \text{ M}\Omega$) as a function of *n*-MCF. **e**, Power outputs of one and two MCF-BFCs composed of the 120-MCF cathode and the 30-GOx/20-MCF anode. In these cases, the two MCF-BFCs were connected with each other in series, and **its power output was also measured with current output through operating a cell potential by an external resistor.**”

On the caption of the revised Supplementary Figure 26b,d: **b**, Power output of the complete BFC **with an external resistor** ($1 \text{ k}\Omega \sim 10 \text{ M}\Omega$) consisting of the 30-GOx/20-MCF anode and 20-MCF cathode as a function of voltage. **d**, Relative power retention (P/P_0) of the complete MCF-BFC in 10 mmol l^{-1} glucose buffer during 35 days. In this case, the power output changes of MCF-BFCs **with external resistors** were continuously measured as a function of

time.

On the caption of the revised Supplementary Figure 29: “Characteristics of the MCF-BFCs by measuring the current output through **external resistors** (in the range of 1 kΩ ~ 10 MΩ) to control cell potential.”

On page 19 in the revised manuscript: “All measurements were carried out in an electrochemical cell **in batch mode**, 50 mL PBS buffer (20 mmol l⁻¹ phosphate and 0.14 mol l⁻¹ NaCl, pH 7.4) at 36.5 °C without stirring.”

2-2. With regard to the reproducibility of the data (error bars).

Response 2-2: We appreciate the reviewer #2’s comment. In the present work, relative voltage retention (%) and relative power retention (%) of the complete MCF-BFCs were obtained from reproducible average values of three to five replicated samples in each experiment conditions.

For more clarifying this issue raised by reviewer #2, we have added the error bars to the calculation of the relative standard deviation. Supplementary Figure 26c and 26d have been now modified, and the caption of the Supplementary Figure 26 has been modified as follows:

On the revised Supplementary Information:

Supplementary Figure 26.

On the caption of Supplementary Figure 26 in the revised Supplementary Information:
 “The error bars show the standard deviation from the mean value of power densities for three to five independent experiments (b, c, d).”

REVIEWERS' COMMENTS:

Reviewer #2 (Remarks to the Author):

The Authors have addressed all the Reviewer's comments.